# Think Before You Retrieve: Learning Test-Time Adaptive Search with Small Language Models

## Abstract

Effective information retrieval requires reasoning over partial evidence and refining strategies as information emerges. Yet current approaches fall short: neural retrievers lack reasoning capabilities, large language models (LLMs) provide semantic depth but at prohibitive cost, and query rewriting or decomposition limits improvement to static transformations. As a result, existing methods fail to capture the iterative dynamics of exploration, feedback, and revision that complex user queries demand. We introduce Orion, a training framework that enables compact models (350M-1.2B parameters) to perform iterative retrieval through learned search strategies. Orion combines: (1) synthetic trajectory generation and supervised fine-tuning to encourage diverse exploration patterns in models, (2) reinforcement learning (RL) that rewards effective query refinement and backtracking behaviors, and (3) inference-time beam search algorithms that exploit the self-reflection capabilities learned during RL. Despite using only 3% of the training data available, our 1.2B model achieves 77.6% success on SciFact (vs. 72.6% for prior retrievers), 25.2% on BRIGHT (vs. 22.1%), 63.2% on NFCorpus (vs. 57.8%), and remains competitive on FEVER, HotpotQA, and MSMarco. It outperforms retrievers up to 200-400× larger on five of six benchmarks. These findings suggest that retrieval performance can emerge from learned strategies, not just model scale, when models are trained to search, reflect, and revise.

## 1 Introduction

Information retrieval has traditionally been framed as a one-shot task: given a query, return the most relevant documents. This formulation assumes that a query fully specifies the user's information need and that relevance can be resolved in a single pass over the corpus (Thakur et al., 2021). Modern neural retrievers have advanced this paradigm significantly (Wang et al., 2020; Karpukhin et al., 2020; Shao et al., 2025; Das et al., 2025; Akkalyoncu Yilmaz et al., 2019; Chen et al., 2024), learning sophisticated representations that encode semantic similarity beyond lexical overlap, achieving strong performance on classic retrieval benchmarks.

However, this one-shot assumption breaks down for complex information needs that require multi-hop reasoning or exploratory search. Current solutions either attempt query reformulation (Yan et al., 2025; Wang et al., 2023), i.e., enriching queries with anticipated evidence, or decomposition (Ammann et al., 2025; Fu et al., 2021), i.e., breaking questions into sub-queries. Both strategies commit to a search plan before observing corpus evidence. When decomposition misses key entities (the "lost-in-retrieval" problem (Zhu et al., 2025b)) or expansion drifts from corpus vocabulary, no recovery mechanism exists. In particular, Tang et al. (2021) showed the severity: models that answer multi-hop questions correctly still fail on 50-60% of the constituent sub-questions.

Interactive retrieval methods (Trivedi et al., 2023; Press et al., 2023; Zhu et al., 2025b; Gao et al., 2025) address this by interleaving retrieval and reasoning, showing that adaptive loops outperform static pipelines. However, these retrieval-augmented generation (RAG) systems place adaptivity in the reasoning layer (LLM controller or generator) rather than in the retrieval layer. The retriever itself remains static, invoked repeatedly but never trained to adapt its search strategy. This overlooks a key point: *the retrieval policy is as important as the reasoning policy*. Deciding how to refine

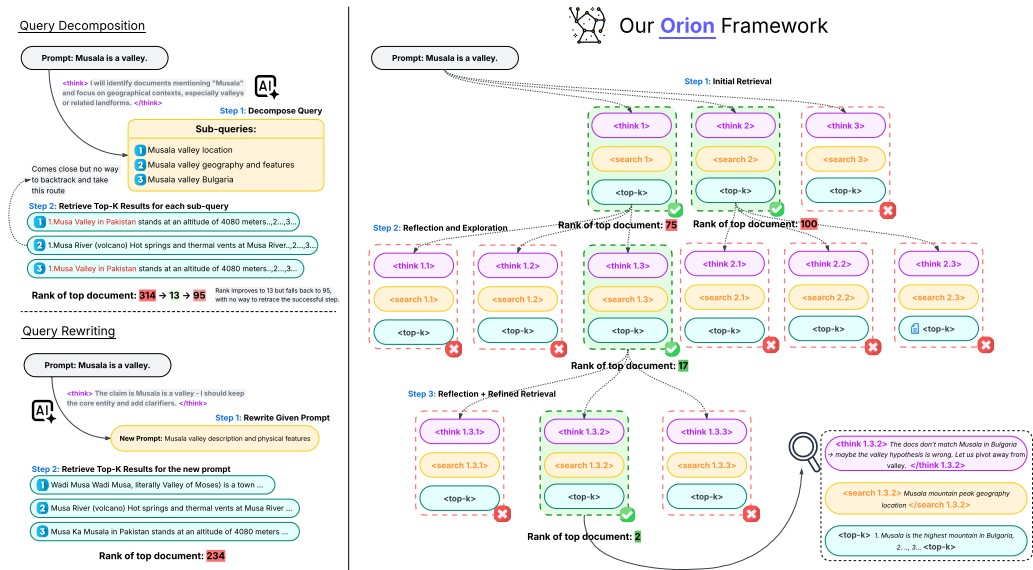

Figure 1: Overview of *Orion*. We illustrate two established query reformulation baselines alongside our proposed Orion framework on an example from the FEVER dataset. While query decomposition fails without corpus feedback and query rewriting yields static reformulations that ignore retrieval results, Orion performs tree-based exploration with structured reasoning spans, revising its strategy as it incorporates contradictory evidence and shifts from valley- to mountain-focused queries-effectively backtracking, refining, and exploring to recover relevant evidence.

queries, explore alternatives, or backtrack from failures is part of retrieval, not just generation. At the same time, scaling analyses (Portes et al., 2025; Zeng et al., 2025) show that retrieval performance grows predictably with model size and pretraining compute. Yet these gains mainly reflect stronger single-shot matching through better embeddings and in-context learning, they do not yield adaptive search policies. Neither costly LLM controllers nor stronger embeddings resolve the core limitation: retrieval models lack the ability to adapt their search strategy in response to observed evidence.

We introduce a different approach: making the retriever itself adaptive. We call this paradigm **test-time adaptive search** and present Orion, a training framework that enables compact models (350M-1.2B parameters) to learn dynamic search policies through synthetic trajectory supervision and turn-level reinforcement learning. Unlike systems that rely on LLMs for test-time reasoning (Jin et al., 2025) or enhance static retrievers with reasoning-aware training (Shao et al., 2025; Das et al., 2025), we train models to internalize diverse search strategies: when to explore alternatives, when to refine promising directions, and when to backtrack from failures.

A key innovation is our turn-level reward structure that leverages standard IR metrics to provide dense feedback at each search step rather than sparse outcome-only signals. This enables models to learn when and how to backtrack from unproductive search directions, a capability that conventional outcome-based training fails to capture. Moreover, our inference algorithm employs beam search with explicit structural markers (`<think>`, `<search_query>`, `<top_k_response>`) that cleanly separate reasoning from querying, keeping search queries concise while allowing thinking spans to incorporate rich retrieval feedback across turns. As Figure 1 illustrates, this design enables systematic exploration of multiple search paths with strategic backtracking when initial directions prove unproductive—moving from failed decomposition attempts to successful evidence recovery through adaptive query reformulation.

Empirically, we demonstrate that compact models can achieve substantial retrieval performance when trained with adaptive search strategies. Despite using models up to 400× smaller, Orion achieves 25.2% nDCG@10 on BRIGHT and 77.6% on SciFact, outperforming both pre-trained and instruction-tuned models, including specialized 3B query rewriting systems. These gains emerge not from stronger embeddings or larger scale, but from learned adaptive behavior: recognizing when

queries fail, exploring alternatives systematically, and recovering from unproductive search paths. Our findings suggest that retrieval intelligence may depend more on learning appropriate search behaviors than on model scale alone, illuminating the potential for compact models to achieve strong performance through targeted training on the core principles of adaptive search.

## 2 RELATED WORK

**Conventional Retrievals**  Classical IR pipelines, lexical metrics and one-pass dense retrieval, optimize a single query, produce top-k hop, and assume that the right evidence is surfaceable with a static query (Voorhees & Tice, 2000; Chen et al., 2017; Maia et al., 2018; Hasibi et al., 2017; Wang et al., 2024; Zhao et al., 2024; Karpukhin et al., 2020; Akkalyoncu Yilmaz et al., 2019). While recent improvements teach retrievers to better follow task intent or ranking signals (Zhuang et al., 2025; Zhang et al., 2025a; Kim & Diaz, 2025; Ko et al., 2025; Rathee et al., 2025), these methods still commit early to a single retrieval state: they neither plan, backtrack, nor adapt the retrieval policy within a session when initial evidence is off-manifold. In high-ambiguity settings, this "fire-and-forget" assumption turns errors in the first hop into answer-level failures. This motivates our approach to introduce an iterative, policy-driven search that can revise hypotheses mid-trajectory.

**Reasoning-based Retrievers and Query Rewriting**  A complementary line trains the retriever to favor evidence that supports multi-step reasoning rather than shallow matches. For example, Shao et al. (2025) builds hard negatives and challenging queries per document, while Das et al. (2025) synthesizes reasoning-conditioned relevance from chain-of-thought traces; both improve reasoning-heavy IR and help downstream RAG. Listwise or pointwise rerankers (Liu et al., 2025; Fan et al., 2025a) push similar supervision during training and then run think-free at inference. On the query side, systems learn to rewrite underspecified or conversational queries directly against retrieval feedback, and some transform documents to produce retrieval-friendly views (Zhu et al., 2025a; Ko et al., 2025; Qin et al., 2025; Yadav et al., 2025; Cha et al., 2025). These advances reduce intent mismatch, but remain single-turn: after one rewrite or rerank the loop stops. Our work closes this by making retrieval itself a multi-turn control policy with turn-level rewards that couple what the model thinks to what it searches next.

**Agentic Retrieval**  A fast-growing body of "agentic" systems trains LLMs to reason while they search: the model alternates thinking, issuing tool calls, reading results, and updating plans. Outcome-rewarded agents (Jin et al., 2025; Jiang et al., 2025) show large gains by letting the model decide when to search and how to reformulate queries against live engines, but can overfit to reward sparsity or exploit quirks of real search APIs. Process- and critic-guided variants (Chang et al., 2024; Dong et al., 2025), retrieval-within-context exemplars (Wang et al., 2025), and latent steering (Xin et al., 2025) inject intermediate guidance and filtering to stabilize trajectories. Other works scale the loop to "deep research" (Li et al., 2025c; Wu et al., 2025; Zheng et al., 2025), or restructure the loop with explicit refine-steps (Shi et al., 2025; Sun et al., 2025a; Peng & Wei, 2025), simple data-centric SFT over realistic web traces (Sun et al., 2025c), or unifying frameworks that couple reasoning and retrieval with curriculum or hybrid knowledge access (Li et al., 2025b;a). Several papers reduce reliance on expensive live search, either by simulating search during training (Sun et al., 2025b; Fan et al., 2025b) or by formalizing the loop as information-foraging over evolving "scent" (Qian & Liu, 2025). Despite their breadth, these agents typically target general QA or open-web tasks, emphasize outcome accuracy over retrieval-policy competence, and require long, brittle trajectories and heavyweight LLM backbones. In contrast, we formulate multi-turn retrieval as a compact, behavior-shaped policy problem and show that small models learn to plan retrieval under turn budgets using group-relative preference optimization (GRPO).

**Reward Design for Multi-Turn IR**  Outcome-only rewards (answer correctness) are simple but sparse, while process-level signals for query quality, evidence selection, and refinement provide denser credit and more stable learning (Zhang et al., 2025b). Recent work adds process feedback (Peng & Wei, 2025; Sun et al., 2025a), lightweight critics (Li et al., 2025a), and gym-style evaluations (Xiong et al., 2025) to reduce over-querying and tool misuse, but most supervise generation or tool use rather than the retrieval step itself. We instead attach rewards directly to each retrieval turn and regularize exploration with structured behaviors, yielding short, predictable plans that continue to improve beyond the first hop, where single-shot retrieval and one-off rewriting struggle.

Our contribution is domain-agnostic: we cast multi-turn search as a behavioral policy that reasons first and then retrieves, and optimize small (350M-1.2B) models with turn-level retrieval rewards. This directly addresses gaps left by single-shot IR (no replanning), reasoning-aware retrievers (no closed-loop search), and agentic RAG (long, costly, outcome-dominated training), and empirically supports iterative planning under tight budgets.

## 3 THE ORION FRAMEWORK

### 3.1 PROBLEM FORMULATION: SEARCH AS A REINFORCEMENT LEARNING TASK

We formalize multi-turn retrieval as a sequential process. Given an initial user query $q_0$ and a document corpus, traditional retrieval produces a single ranking. *Test-time adaptive search* extends this by generating a sequence of search actions, where each step refines the query based on observed evidence.

At turn $t$ (analogous to a time step in RL), the environment state is $s_t = (q_0, H_t)$, where the history $H_t = \{(\phi_i, q_i, \mathcal{R}(q_i))\}_{i=1}^{t-1}$ records prior interactions. Here, $\phi_i$ is the reasoning trace, $q_i$ the issued query, and $\mathcal{R}$ the deterministic retriever that returns documents for a given query.

Each action $a_t$ consists of two parts: the reasoning step $\phi_t$ and the refined query $q_t$. The policy first generates reasoning from the current state, $\phi_t \sim \pi_\theta(\cdot \mid s_t)$, and then refines the query, $q_t \sim \pi_\theta(\cdot \mid s_t, \phi_t)$. The history is updated as $H_{t+1} = H_t \cup \{(\phi_t, q_t, \mathcal{R}(q_t))\}$. To guide generation, we delimit components using the structural tokens shown in Table 1.

Table 1: Structural tokens used to delimit reasoning, queries, and retrieval responses.

| Token | Purpose |
|---|---|
| `<user_query>...</user_query>` | Original query $q_0$ (fixed) |
| `<think>...</think>` | Reasoning trace $\phi_t$ |
| `<search_query>...</search_query>` | Refined queries $q_t$ ($t > 0$) submitted to $\mathcal{R}$ |
| `<top_k_response>...</top_k_response>` | Retrieved results $\mathcal{R}(q_t)$ |

An episode terminates when the target document appears in the top-$k$ results or when the maximum number of turns $T_{\max} = 5$ is reached. We train LLM policies $\pi_\theta$ with parameters $\theta$ to maximize expected retrieval success through turn-level RL, with KL regularization against a reference policy $\pi_{\text{ref}}$.

### 3.2 SYNTHETIC TRAJECTORY GENERATION

One challenge in training retrieval models is the mismatch between benchmarks dominated by short, factoid-style queries (Nguyen et al., 2016; Thakur et al., 2021) and real-world search demands that require multi-step reasoning, reformulation, and hypothesis testing. While ReasonIR (Shao et al., 2025) addressed part of this gap by generating longer queries and showing that decomposition degrades performance, we view synthetic data generation as a way of teaching models *how to search*: treating retrieval as a process that unfolds through cycles of reasoning, querying, and refinement.

We model multi-turn search through diverse behavioral strategies that capture various search patterns, motivated by prior findings that diversity, rather than scale alone, is key for generalization (Jung et al., 2025; Wen et al., 2025). Following established approaches in query reformulation (Diaz, 2016; Balaneshin-kordan & Kotov, 2016), we treat queries as nodes in a reformulation graph where each node can spawn alternative search directions. This framework allows us to synthesize behavioral archetypes such as breadth-first and depth-first traversal, evidence-driven reformulation, stochastic wandering, and trajectory-aware strategies like recognizing early success or reflecting on failure. To capture these patterns, we construct an ultra-feedback pool (Cui et al., 2024) of multi-turn search traces generated by eight popular LLMs on the training splits of several retrieval datasets, ensuring robustness against model-specific biases while sampling structured think-query-retrieve cycles that preserve diversity and coherent reasoning flows. Further details are provided in Appendix B.2, which discuss in detail the different data generation algorithms and ultra-feedback models.

Additionally, we explore whether models benefit from structured exposure to search behaviors through curriculum learning during SFT, where training progresses from simple reformulation strategies to complex multi-hypothesis coordination, or whether random presentation of diverse behavioral patterns proves equally effective. We also examine individual algorithm training to isolate the contribution of specific search behaviors, and employ model souping techniques inspired by SmolLM3 and Llama-Nemotron-Super's approach, which uses MergeKit to combine behavioral specialists with exponential weighting that favors sophisticated strategies (Bakouch et al., 2025; Goddard et al., 2024; Bercovich et al., 2025). These comparisons aim to reveal how models best internalize the spectrum of search capabilities encoded in our synthetic data.

## 3.3 Training Framework

Our training consists of two stages: SFT establishes multi-turn search scaffolding, followed by GRPO (Shao et al., 2024) that refines search behavior through turn-level rewards.

**Supervised Fine-Tuning**  We perform supervised fine-tuning on the synthetic dataset to establish the structural framework of multi-turn search. Each training example includes explicit markers for reasoning, query emission, and retrieval results, ensuring the model learns to generate well-formed cycles of `<think>`, `<search_query>`, and `<top_k_response>` tokens in this order. This stage grounds the model in the format and temporal flow of iterative search traces, aligning internal reasoning with external retrieval actions across multiple turns. Unlike conventional fine-tuning on single-turn queries, it establishes the behavioral foundation for subsequent training with GRPO.

**Group-Relative Policy Optimization**  Building on this initialization, we apply GRPO to refine search behavior with retrieval-based feedback. At each turn, the model generates multiple reasoning-query candidates, which are executed against the retriever and scored. Each generation includes a "think" segment followed by a search query. For each query, we compute a reward that combines two normalized components: (i) the cosine similarity of the retrieved document to the query and (ii) the rank of the best-matching document in the corpus. Similarity scores are normalized to $[0, 1]$ by mapping negative values as $(\text{sim} + 1)/2$, while rank is normalized as $1 - \text{rank}/|C|$, where $|C|$ is the size of the corpus. Each signal contributes equally to the reward.

From $G$ sampled generations per turn (where $G$ denotes the group size), the highest-reward candidate is selected to advance the context for the next turn. While context advancement is greedy at each turn, GRPO training incorporates all candidates through group-relative policy updates. Advantages are normalized relative to the candidate set, and policy gradients are computed using all candidates, not just the highest-reward one. KL regularization against a reference model stabilizes language generation, while group size and horizon (both set to 4) govern exploration depth. The full algorithm is provided in Appendix B.3.

## 3.4 Inference with Orion

Drawing inspiration from DeepConf's confidence-based filtering (Fu et al., 2025) and graph-based reformulation techniques (Diaz, 2016), Orion leverages the enhanced self-reflection capabilities developed through our SFT and GRPO training stages to perform adaptive search via structured beam management. While DeepConf computes token confidence as $C_i = -\frac{1}{k}\sum_{j=1}^{k} \log P_i(j)$ over top-$k$ token probabilities, Orion evaluates retrieval effectiveness through structured relevance assessment. For each beam thread containing the complete context history ending with `<think>`, `<search_query>`, and `<top_k_response>`, we prompt the model with a new follow-up think-reflection: *"Given turn $t$ and search query $q_t$, the retrieved documents are relevant to the user query $\{q_0\}$."*. We then compute perplexity as $\text{PPL} = \exp\left(-\frac{1}{N}\sum_{i=1}^{N} \log P(x_i|x_{<i})\right)$ on the model's relevance judgment. This approach captures learned metacognitive assessment of search quality rather than surface-level token uncertainty, providing a semantically grounded confidence signal for beam ranking. As detailed in Algorithm 1, the method maintains a search tree where each query expands into $M$ candidate branches, with $M$ denoting the number of alternatives pursued per beam. Each node then generates $M$ children, after which survival-based pruning retains the top $B$ candidates, where $B$ is the beam size controlling how many hypotheses survive to the next step. Running

---

**Algorithm 1** Orion-Beam-Search for Test-Time Adaptive Retrieval

---

**Require:** User query $q_0$, policy $\pi_\theta$, retriever $\mathcal{R}$, beam size $B$, expansion width $M$, max turns $T_{\max}$
1: Initialize active beams $\mathcal{B}_0 = \{(q_0, \emptyset, 0)\}$, where each beam is $(\text{query}, \text{history}, \text{confidence})$
2: **for** $t = 1$ to $T_{\max}$ **do**
3:     $\mathcal{B}_{\text{candidates}} \leftarrow \emptyset$
4:     **for** each beam $(q_i, H_i, c_i) \in \mathcal{B}_{t-1}$ **do**
5:         Sample $M$ reasoning-query pairs: $\{(\phi_{i,j}, q_{i,j})\}_{j=1}^M \sim \pi_\theta(\cdot|q_i, H_i)$
6:         **for** $j = 1$ to $M$ **do**
7:             Execute retrieval: $r_{i,j} \leftarrow \mathcal{R}(q_{i,j})$
8:             Construct context: $\text{ctx}_{i,j} \leftarrow H_i \cup \{(\phi_{i,j}, q_{i,j}, r_{i,j})\}$
9:             Generate relevance prompt: $p \leftarrow$ `"Given turn {t} and search query {q`$_t$`},` `the retrieved documents are relevant to the user query` `{q`$_0$`}."`
10:             Compute perplexity: $\text{ppl}_{i,j} \leftarrow \exp\left(-\frac{1}{N}\sum_{k=1}^N \log \pi_\theta(y_k|y_{<k}, \text{ctx}_{i,j}, p)\right)$
11:             Add candidate: $\mathcal{B}_{\text{candidates}} \leftarrow \mathcal{B}_{\text{candidates}} \cup \{(q_{i,j}, \text{ctx}_{i,j}, \text{ppl}_{i,j}^{-1})\}$
12:         **end for**
13:     **end for**
14:     Sort candidates: $\mathcal{B}_{\text{sorted}} \leftarrow \{b \in \mathcal{B}_{\text{candidates}} : c(b_1) \geq c(b_2) \geq \ldots \geq c(b_n)\}$
15:     Select survivors: $\mathcal{B}_t \leftarrow \text{top-}B(\mathcal{B}_{\text{sorted}})$
16:     **if** any beam achieves retrieval success **then**
17:         **return** best beam from $\mathcal{B}_t$
18:     **end if**
19: **end for**
20: **return** highest confidence beam from $\mathcal{B}_{T_{\max}}$

---

inference in this way balances exploration of diverse alternatives with focused refinement on the model-perceived most promising trajectories.

## 4 EXPERIMENTS

### 4.1 EVALUATION SETUP

**Benchmarks** We evaluate on five datasets that reflect different aspects of search complexity. From the BEIR benchmark (Thakur et al., 2021), single-hop IR is measured on NFCorpus (biomedical retrieval), while multi-hop tasks include FEVER and SciFact (fact-checking) and HotpotQA (question answering). We also use BRIGHT (SU et al., 2025), a dataset of reasoning-intensive queries from domains such as economics, mathematics, and programming that require deeper analysis to identify relevant documents. Because our models are not explicitly trained on such reasoning-heavy tasks, BRIGHT tests whether the learned search strategies generalize beyond the training distribution and adapt to more challenging retrieval settings.

**Metrics** We evaluate retrieval effectiveness using nDCG@k (ranking quality with graded relevance), Success@k (whether target documents appear in the top-k results), Recall@K (proportion of relevant documents retrieved in the top-k results), and MRR (mean reciprocal rank emphasizing early precision). Together, these metrics capture both effectiveness and efficiency.

### 4.2 MODELS AND BASELINES

**Orion Models** We build on the LFM2 architecture with 350M, 700M, and 1.2B parameter variants. LFM2's hybrid design, combining multiplicative gates with short convolutions, delivers significant inference speed gains over standard transformers, making it well-suited for production retrieval systems (Liquid AI, 2025a;b). The smaller parameter counts let us test whether learned search strategies can compensate for reduced model scale.

**Baselines** We compare against three categories of systems. First, general-purpose instruction-tuned LLMs, including models from the GPT, Llama, and Qwen families. These represent the costly test-time reasoning approaches that Orion is designed to replace. Second, traditional IR baselines such as BM25 and dense retrieval with MiniLM-L6-v2 embeddings. We deliberately use the compact MiniLM (all-MiniLM-L6-v2; 22.7M parameters) as the retrieval backend to create challenging conditions where learned strategies must compensate for weaker embeddings (Sun et al., 2024), underscoring the practical value of our approach. Third, state-of-the-art baselines such as DeepRetrieval (Jiang et al., 2025), which introduces a 3B-parameter model for relevant query generation. Additional discussion of baseline choices is provided in Appendix A.

## 4.3 RESULTS

**Classic information retrieval benchmarks.** On BEIR tasks (Table 2, nDCG@10), we observe distinct performance patterns across fact-checking scenarios. For scientific verification like SciFact, our approach performs competitively with baselines, while on FEVER we achieve comparable results to LLMs but trail specialized retrievers like DeepRetrieval (84.1%) and BM25 dense (82.5%). This divide suggests that learned search strategies show benefits when domain expertise is required, though lexical matching remains effective. Our models also show consistent performance on biomedical tasks like NFCorpus where traditional dense methods struggle.

**Reasoning-intensive retrieval tasks.** BRIGHT (Table 3) shows varied performance patterns across reasoning domains. Our overall average compares favorably to baselines, with notable performance on coding tasks where we achieve 32.9% on Pony while trailing BM25 dense on LeetCode. We observe consistent results across the theorem-based category, with Orion-Medium achieving the highest score on AoPS, despite no math or coding

Table 2: **nDCG@10 scores (%) across *classic* information retrieval tasks from the BEIR benchmark.** Standard deviations are omitted as they are negligible (often zero). The best score on each dataset is shown in bold. *Parameter counts approximated by Abacha et al. 2025; †values imported from the original paper due to unavailable model checkpoints.

| Model/Retriever | Multi-Hop | | | Single-Hop |
|---|---|---|---|---|
| | FEVER | HotpotQA | SciFact | NFCorpus |
| *General-Purpose LLMs* | | | | |
| GPT-4.1 | 61.3 | **74.8** | 72.4 | 57.8 |
| GPT-4.1-mini | 58.8 | 71.3 | 72.3 | 56.5 |
| GPT-4o (200B*) | 59.5 | 73.6 | 70.8 | 55.8 |
| GPT-4o-mini (8B*) | 54.9 | 68.6 | 69.8 | 53.7 |
| Llama 3.1-405B | 64.8 | 73.8 | 70.3 | 56.2 |
| Llama 3.1-70B | 63.4 | 72.7 | 70.7 | 56.5 |
| Llama 3.1-8B | 59.7 | 68.1 | 70.2 | 55.3 |
| Llama-3.2-3B | 57.6 | 66.6 | 67.1 | 55.0 |
| Qwen3-235B | 60.2 | 72.5 | 72.6 | 57.0 |
| Qwen2.5-7B | 56.7 | 66.8 | 69.2 | 55.1 |
| Qwen2.5-3B | 54.7 | 64.8 | 67.2 | 56.3 |
| *Retrieval Baselines* | | | | |
| BM25 (dense) | 82.5 | 70.0 | 64.5 | 37.0 |
| BM25 (sparse) | 44.2 | 61.1 | 57.3 | 14.7 |
| MiniLM-L6-v2 (22.7M) | 42.5 | 48.7 | 50.5 | 39.9 |
| DeepRetrieval† (3B) | **84.1** | 70.1 | 66.4 | 37.7 |
| *Orion Models (ours)* | | | | |
| Orion-Large(1.2B) | 65.3 | 71.6 | **77.6** | **63.2** |
| Orion-Medium(700M) | 63.3 | 68.5 | 71.1 | 60.5 |
| Orion-Small(350M) | 57.7 | 64.1 | 70.9 | 60.5 |

training data. These results suggest that learned search strategies transfer across complex reasoning scenarios.

## 5 DISCUSSION

**Does RL help beyond SFT?** A central question for Orion is whether RL delivers benefits that go beyond what SFT alone provides. This is most clearly illustrated on BRIGHT, where base retrievers perform poorly, underscoring that static embedding similarity cannot sustain multi-turn search (Table 4). SFT nearly doubles performance by teaching models to produce structured `<think>` and `<search_query>` sequences, but this scaffolding primarily induces mechanistic turn-taking without real strategy: once an initial trajectory goes astray, the model rarely recovers (Figure 2). Adding GRPO produces only modest topline gains (Table 4), yet

Table 4: **Multi-turn search performance of Orion-Large on BRIGHT** (nDCG@10). SFT nearly doubles performance over Base (+9.2–14.8%), while GRPO yields only modest additional gains (+1–2%).

| | *Orion Models* | | |
|---|---|---|---|
| Method | 1.2B | 700M | 350M |
| Base | 0.104 | 0.098 | 0.062 |
| SFT | 0.207 | 0.195 | 0.154 |
| GRPO | 0.212 | 0.199 | 0.156 |

Table 3: **nDCG@10 scores (%) on *reasoning-intensive* retrieval tasks from the BRIGHT benchmark:** biology (Bio.), earth science (Earth.), economics (Econ.), psychology (Psy.), robotics (Rob.), stack overflow (Stack.), sustainable living (Sus.), LeetCode (Leet.), Pony, AoPS, TheoremQA with question retrieval (TheoQ.) and with theorem retrieval (TheoT.). "Avg." denotes the macro average score across 12 datasets. Standard deviations are omitted as they are negligible (often zero). The best score on each dataset is shown in bold. *Parameter counts approximated by Abacha et al. 2025.

| | StackExchange | | | | | | | Coding | | Theorem-based | | | |
| Model/Retriever | Bio. | Earth. | Econ. | Psy. | Rob. | Stack. | Sus. | Leet. | Pony | AoPS | TheoQ. | TheoT. | Avg. |
|---|---|---|---|---|---|---|---|---|---|---|---|---|---|
| | *General-Purpose LLMs* | | | | | | | | | | | | |
| GPT-4.1 | 39.7 | 40.9 | 24.9 | **33.2** | 17.9 | 14.9 | 29.1 | 20.1 | 14.9 | 4.4 | 19.9 | 7.8 | 22.1 |
| GPT-4.1-mini | 38.0 | 38.4 | **26.2** | 32.8 | 18.6 | 14.0 | 29.1 | 21.2 | 10.2 | 4.1 | 19.9 | 7.3 | 21.2 |
| GPT-4o (200B*) | 30.4 | 35.6 | 20.7 | 30.3 | 16.6 | 11.5 | 22.0 | 18.9 | 14.5 | 2.6 | 10.7 | 5.7 | 18.3 |
| GPT-4o-mini (8B*) | 27.0 | 31.7 | 16.7 | 31.7 | 14.4 | 13.3 | 26.1 | 17.1 | 10.8 | 3.6 | 10.3 | 3.2 | 16.7 |
| Llama 3.1-405B | 36.6 | 35.1 | 21.3 | 29.6 | 14.0 | 12.4 | 22.6 | 18.2 | 6.2 | 1.1 | 13.2 | 2.5 | 18.3 |
| Llama 3.1-70B | 32.6 | 36.8 | 21.8 | 29.7 | 16.1 | 13.7 | 24.2 | 21.3 | 5.5 | 1.1 | 14.5 | 3.2 | 18.0 |
| Llama 3.1-8B | 32.2 | 32.4 | 21.7 | 26.7 | 15.5 | 10.6 | 22.3 | 15.0 | 5.1 | 2.3 | 10.7 | 1.4 | 16.7 |
| Llama-3.2-3B | 23.4 | 28.2 | 16.8 | 23.2 | 12.0 | 10.3 | 18.6 | 13.8 | 4.8 | 0.7 | 6.7 | 0.7 | 13.6 |
| Qwen3-235B | **43.4** | 38.8 | 22.8 | 33.1 | **20.4** | 14.9 | 30.4 | 18.0 | 12.1 | 3.7 | 18.8 | 7.0 | 21.8 |
| Qwen2.5-7B | 25.6 | 26.4 | 15.8 | 25.4 | 11.7 | 10.5 | 21.4 | 16.4 | 10.2 | 2.1 | 10.5 | 4.4 | 15.4 |
| Qwen2.5-3B | 22.0 | 25.6 | 14.5 | 21.7 | 11.7 | 12.3 | 16.8 | 17.6 | 5.9 | 2.2 | 10.8 | 1.3 | 13.5 |
| | *Retrieval Baselines* | | | | | | | | | | | | |
| BM25 (dense) | 18.1 | 27.5 | 15.7 | 12.6 | 13.2 | 19.3 | 15.2 | **24.2** | 7.7 | 6.5 | 10.4 | 4.8 | 14.6 |
| BM25 (sparse) | 7.7 | 14.1 | 10.3 | 6.3 | 9.7 | 9.4 | 9.1 | 12.8 | 0.4 | 1.0 | 2.8 | 0.0 | 7.0 |
| MiniLM-L6-v2 (22.7M) | 16.7 | 20.5 | 11.6 | 12.1 | 12.3 | 7.8 | 14.0 | 20.9 | 1.5 | 2.7 | 6.5 | 0.5 | 11.1 |
| | *Orion Models (ours)* | | | | | | | | | | | | |
| Orion-Large (1.2B) | 37.8 | **41.8** | 23.5 | 26.8 | 18.5 | **21.7** | **31.5** | 23.2 | **32.9** | 5.8 | **25.9** | **13.3** | **25.2** |
| Orion-Medium (700M) | 33.9 | 39.4 | 25.1 | 26.7 | 19.6 | 20.9 | 26.9 | 23.3 | 30.9 | **7.3** | 25.4 | 11.4 | 24.2 |
| Orion-Small (350M) | 31.3 | 34.3 | 21.1 | 25.7 | 19.0 | 22.2 | 24.2 | 16.8 | 24.4 | 5.7 | 20.1 | 9.8 | 21.2 |

these small improvements mask more significant behavioral shifts. Recall rises more noticeably (See Table 9), reflecting broader coverage of relevant documents and more stable trajectories across runs. Most crucially, RL induces backtracking behavior: as shown in Figure 2, Orion-Large with GRPO exhibits a drastic increase in backtracking compared to Orion-SFT. This echoes recent findings that RL often imparts capabilities rather than large static metric gains (Shao et al., 2024). In Orion, the induced capability is adaptive recovery, that is, knowing when and how to pivot during multi-turn search. While SFT provides the scaffolding, RL equips models with the strategic ability to use it effectively.

**Does Behavioral Diversity in Synthetic Data Matter?** Here, We ask what kinds of behaviors should be encoded in synthetic trajectories if models are to acquire effective search strategies. Cross-dataset comparisons reveal that no single algorithm consistently dominates (Tables 7, 8). Instead, effectiveness is tightly coupled to task structure. On BRIGHT, where reasoning errors quickly cascade, recovery mechanisms such as backtracking and validation are indispensable (Table 5). In contrast, multi-hop fact verification tasks like FEVER & HotpotQA benefit from exploratory behaviors that uncover complementary evidence spans, more granular results are presented in Appendix D.3.

Table 5: nDCG@10 performance of different search behaviours on BRIGHT. Bold values indicate the best-performing behaviour.

| Algorithm | BRIGHT |
|---|---|
| Early-Success Validator | **0.175** |
| Wrong-Direction Specialist | 0.173 |
| Greedy Hill Climber | 0.169 |
| Best-First Hypothesis Selector | 0.168 |
| Exploitation-Heavy Validator | 0.166 |
| Depth-First Driller | 0.166 |
| Multi-Beam Parallel | 0.149 |
| Adaptive Context Learner | 0.140 |
| Random Walk Wanderer | 0.140 |
| Breadth-First Explorer | 0.113 |

Taken together, these results underscore a central insight: robust retrieval competence does not arise from mastering a single tactic, but from the ability to orchestrate exploration, exploitation, and recovery as complementary tools, deploying each in contextually appropriate ways. In line with classic IR theory, the "right" behavior is not universal but contingent on the evidence landscape and error tolerance of each benchmark (Vakkari, 1999; 2001; Sutcliffe & Ennis, 1998). Our findings extend this principle to multi-turn neural search, showing that adaptability across behavioral archetypes is itself the capacity that models must learn.

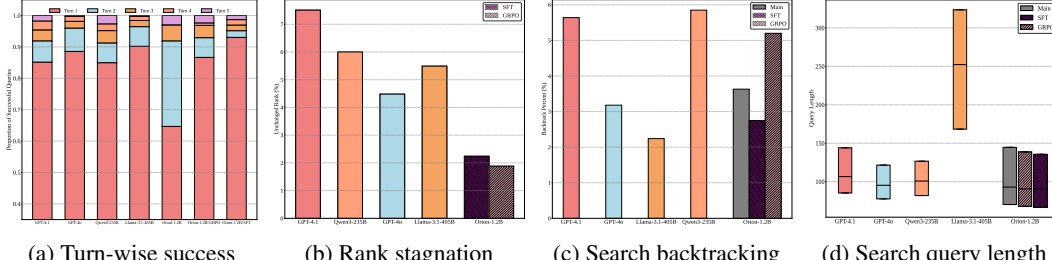

| (a) Turn-wise success | (b) Rank stagnation | (c) Search backtracking | (d) Search query length |

Figure 2: We present further behavioral analysis of Orion-Large on BRIGHT: (a) demonstrates how successful queries distribute across search turns for different models, while (b) illustrates the proportion of queries with unchanged rankings across turns, indicating repetitive search patterns and the inability to overcome search stagnation. (c) measures backtracking behavior by counting queries where rankings ($r$) deteriorate then recover ($r_{i-1} > r_i < r_{i+1}$), while (d) shows search query length distribution, demonstrating our models generate relatively succinct search queries. Results for Orion-Small and Orion-Medium variants are provided in Appendix D.3.

**Should models learn to fail better in retrieval?** Retrieval intelligence is not only about early success but about failing productively. In manual inspection, we noticed that general-purpose LLMs often fall into "revolving loops," repeating near-identical queries without escape. Orion shares this vulnerability, but more often manages to break free-pivoting with substantive reformulations rather than shallow lexical edits. This qualitative difference shows up quantitatively: proprietary models exhibit high rank stagnation (Figure 2b), while Orion's lower stagnation reflects more active in-the-moment recovery. GRPO further strengthens this behavior by increasing backtracking (Figure 2c), signaling a greater willingness to revisit failed directions when initial searches falter.

Yet failure is a double-edged sword. Recovery that comes too late, or occurs too often, distorts the distribution of successful turns (Figure 2a), with later steps yielding diminishing returns. This coincides with excessive looping but also reflects the inherent difficulty of later turns, underscoring the tension between resilience and efficiency. Similar dynamics appear on FEVER and HotpotQA (Appendix Figure D.3), where Orion again shows higher backtracking and lower stagnation than GPT-4.1 and Qwen2.5. These patterns suggest that effective models must balance recovery capabilities with efficiency, learning to backtrack strategically while avoiding excessive course-correction that impedes overall progress. Future training strategies should target this balance directly, rewarding decisive recovery while penalizing shallow repetition.

## 6 CONCLUSION

In summary, Orion shows that retrieval intelligence is not a function of scale but of strategy. By combining synthetic trajectories, reinforcement learning, and beam search, compact models (350M–1.2B) learn to detect failure, redirect search, and recover systematically—capabilities that emerge from targeted behavioral training rather than massive parameter counts. Despite being hundreds of times smaller than prevailing LLMs, Orion matches or surpasses them across six benchmarks, excelling on reasoning-heavy datasets like BRIGHT. For production systems, this means that reliable, cost-efficient retrieval no longer requires expensive controllers: compact models trained on diverse behaviors suffice. The broader lesson is clear—the future of retrieval lies in models that know how to search, not just in building ever-larger models.

## 7 ETHICS STATEMENT

This work investigates methods for improving information retrieval through adaptive search strategies in compact models. The research does not involve human subjects, sensitive personal data, or deployment in high-risk domains. All datasets used are publicly available retrieval benchmarks (e.g., BEIR, BRIGHT, FEVER, SciFact, HotpotQA, NFCorpus, MS MARCO) that contain curated, non-personal text.

The primary societal benefit of this work is efficiency: Orion demonstrates that strong retrieval performance can be achieved with models several hundred times smaller than existing systems. This reduces energy consumption and compute cost, lowering barriers to research and deployment. It also enables broader access to effective retrieval without requiring reliance on proprietary LLMs.

At the same time, retrieval systems can amplify biases present in training data or surface harmful content. While Orion focuses on adaptive search strategies rather than corpus construction, we acknowledge that our methods inherit dataset biases and limitations. Future work should investigate fairness-aware training objectives and evaluate adaptive retrieval across diverse cultural and linguistic contexts.

Finally, although our experiments are limited to static academic benchmarks, real-world deployment of retrieval models must carefully consider privacy, misinformation risks, and potential misuse. By framing retrieval competence as strategy rather than scale, our work seeks to promote more transparent and resource-conscious directions for information systems.

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

APPENDIX

## A  DIFFERENCE BETWEEN ORION AND PREVIOUS WORK

Orion addresses a fundamentally different challenge from most existing retrieval enhancement methods. While prior work has largely focused on static improvements to retrievers or on leveraging external search infrastructure, we study how compact models can adaptively decide what to search for next during inference. This shift in focus changes the nature of the baselines and the comparisons that matter.

**Reasoning-aware retrievers and rerankers** illustrate this contrast. Approaches such as ReasonIR(Shao et al., 2025), RADER (Das et al., 2025), or Rank1 (Liu et al., 2025) augment retrieval by designing better training objectives, mining harder negatives, or refining rankings over a fixed candidate set. Yet these methods remain bound to a single-shot paradigm: once the initial query is issued, the system has no mechanism to backtrack from failed hypotheses or to explore new search directions. Orion instead operates at a different layer of the problem, the exploration process itself, by modeling how to reformulate queries across turns. These directions are not competitive but complementary: reasoning-aware retrievers improve how queries and documents are encoded, while Orion determines what to search for next. In fact, Orion's strategies can be layered on top of strong retrievers like ReasonIR or RADER, replacing the reliance on costly GPT-4o rewrites for multi-turn search.

A second line of related work involves **systems that rely on production search engines**, such as Search-R1 (Jin et al., 2025). These methods delegate the hardest parts of retrieval to mature

infrastructures, web-scale corpora, continuously updated indexes, and heavily optimized ranking algorithms. Orion is deliberately studied under a more constrained setting: a fixed, offline corpus and a lightweight retriever (MiniLM-L6-v2). Within this environment, success cannot be attributed to superior infrastructure; instead, the model must genuinely learn strategies of multi-step reasoning, hypothesis refinement, and recovery from errors.

The **most direct comparisons** arise against models explicitly designed for iterative information seeking. State-of-the-art agentic systems such as GPT-4.1, GPT-4o, Llama 3.1-3.1-405B, Qwen3-235B serve as natural baselines, as they possess the reasoning capabilities needed to engage in multi-turn retrieval. DeepRetrieval (Jiang et al., 2025) is our closest methodological neighbor. Like Orion, it applies RL to retrieval, confirming that RL can improve search. However, its focus lies on unsupervised query generation via PPO, while Orion models full trajectories, assigns structured rewards at the turn level, and integrates reasoning spans to guide exploration. In this sense, their work validates the importance of adaptive retrieval, but differs in execution.

Finally, the scale comparison underscores Orion's contribution. Despite being only 1.2B parameters, Orion consistently outperforms models 200-400$\times$ larger on five of six benchmarks. For reference, DeepRetrieval reports efficiency gains over GPT-4o with a 3B model. Orion matches or surpasses those results with a model 2.5$\times$ smaller, while directly competing with the largest reasoning-capable systems. Moreover, Orion appears to generalize across out-of-distribution datasets without requiring dataset-specific retraining, in contrast to DeepRetrieval, where separate models are trained for each domain. This efficiency and robustness together demonstrate that adaptive search, when explicitly modeled, can both close the gap to far larger models and broaden applicability across diverse retrieval settings.

# B    DETAILED ORION METHODOLOGY

## B.1    SYNTHETIC TRAJECTORY GENERATION

**Ultra-Feedback Pool Construction**    We collected multi-turn search behaviors from 8 language models (GPT-4.1, GPT-4.1-mini, GPT-4o, GPT-4o-mini, Llama 3.1-3.1-405B, Llama 3.1-8B, Llama 3.2-3B, Qwen2.5-7B) across 4 retrieval datasets (FiQA, HotpotQA, MS Marco, SciFact). Each model performed 5-turn iterative retrieval on the same user queries, creating a diverse pool of search behaviors.

For each user query $q_i$, we obtained from each model $M_j$:

- Search traces (refined queries): $\{q_{i,j,t}\}_{t=1}^{5}$ (search queries per turn)
- Thinking traces: $\{\phi_{i,j,t}\}_{t=1}^{5}$ (reasoning for turn $t$: initial planning for $t = 1$, reflection + planning for $t > 1$)
- Retrieval results: $\{R_{i,j,t}\}_{t=1}^{5}$ (top-$k$ documents retrieved for query $q_{i,j,t}$)
- Performance metrics: $\{\cos_{i,j,t}, \text{rank}_{i,j,t}\}_{t=1}^{5}$ (cosine similarity to target, ground truth rank)

The flow structure is: $\phi_{i,j,t} \rightarrow q_{i,j,t} \rightarrow \mathcal{R}(q_{i,j,t}) \rightarrow \phi_{i,j,t+1} \rightarrow q_{i,j,t+1}$

This creates an ultra-feedback pool $\mathcal{U} = \{(q_0, \{(\phi_{i,j,t}, q_{i,j,t}, R_{i,j,t}, \cos_{i,j,t}, \text{rank}_{i,j,t})\})\}$ containing diverse search patterns for each query. While, we use a unified think-sequence generator, context from ultra-feedback source models and system-prompt minimal edit requirement force the diversity from the original model's reflections & planning to be retained while still producing semantically coherent think sequences on a new thread of search queries.

## B.2 Synthetic Search Behaviors

Below, we summarize the synthetic search behaviors that form the basis of our synthetic data generation process.

Table 6: Synthetic search behaviors used in our framework. Each behavior represents a distinct strategy, spanning systematic exploration (e.g., breadth-first, depth-first), adaptive refinement (e.g., adaptive context learning, hill climbing), and validation approaches (e.g., early-success, exploitation-heavy). Together, these archetypes illustrate how models can navigate, adapt, and coordinate across diverse search pathways.

| Behavior | What It Does | Example |
|---|---|---|
| **Adaptive Context Learner** (Pawar et al., 2016) | Learns from search results and adds relevant keywords from retrieved documents | Query: "climate change" → sees papers mention "carbon emissions" → next query: "climate change carbon emissions" |
| **Random Walk Wanderer** (Pearson, 1905) | Explores randomly in different directions without a clear plan | "solar panels" → "renewable energy" → "wind turbines" → "energy storage" (jumping around topics) |
| **Breadth-First Explorer** (Moore, 1959) | Systematically covers all related topics before going deeper | First: "AI applications", "AI ethics", "AI history" → then dive deeper into each area |
| **Depth-First Driller** (Lucas, 1882) | Goes deep into one direction until exhausted, then backtracks | "machine learning" → "neural networks" → "deep learning" → "transformers" → "attention mechanisms" (keeps drilling down) |
| **Wrong-Direction Specialist** (Ertmer & Newby, 1996) | Recognizes when searches are getting worse and explains why | "Looking for Python tutorials but keep finding snake facts - my query is too ambiguous" |
| **Early-Success Validator** (Haarnoja et al., 2018) | Recognizes good results early and sticks with successful approaches | First query works well → "This is giving me relevant papers, let me refine this direction further" |
| **Exploitation-Heavy Validator** (Even-Dar & Mansour, 2001) | Keeps optimizing successful queries without trying new approaches | Found good results with "deep learning NLP" → keeps refining: "deep learning natural language processing", "deep learning text analysis" |
| **Greedy Hill Climber** (Selman & Gomes, 2006) | Always picks the next query that seems like the biggest improvement | Tests multiple query variations and always picks the one that got the best results |
| **Best-First Hypothesis Selector** (Korf, 1999) | Manages multiple search ideas and picks the most promising one to pursue | Has 3 search directions, evaluates which is working best, focuses on that one |
| **Multi-Beam Parallel** (Steinbiss et al., 1994) | Runs several different search strategies at the same time | Simultaneously searches "climate data", "weather patterns", and "temperature trends" |

## B.3 GRPO ALGORITHMS

We present our GRPO training and reward algorithms in Algorithms 2 and 3 below.

---

**Algorithm 2** GRPO-based Retrieval Training

---

**Require:** Dataset $D$, policy $\pi_\theta$, reference $\pi_{\text{ref}}$, retriever $\mathcal{R}$, group size $G$, horizon $T_{\max}$
1: **for all** $q \in D$ **do**
2:     Initialize history $H_1 \leftarrow \emptyset$
3:     **for** $t = 1$ to $T_{\max}$ **do**
4:         Sample $G$ candidate actions: $(\phi_t^{(i)}, q_t^{(i)}) \sim \pi_\theta(\cdot \mid q_0, H_t)$
5:         **for** $i = 1$ to $G$ **do**
6:            $r_t^{(i)} \leftarrow \mathcal{R}(q_t^{(i)})$
7:            $R^{(i)} \leftarrow \text{reward\_function}(r_t^{(i)})$
8:         **end for**
9:         $A^{(i)} \leftarrow R^{(i)} - \frac{1}{G} \sum_j R^{(j)}$
10:       $\theta \leftarrow \theta - \eta \nabla_\theta \left[ - \frac{1}{G} \sum_i A^{(i)} \log \pi_\theta(a_t^{(i)} \mid q_0, H_t) + \beta D_{\text{KL}}^t(\pi_\theta \| \pi_{\text{ref}}) \right]$
11:       Sample $i^* \propto R^{(i)}$
12:       $H_{t+1} \leftarrow H_t \cup \{(\phi_t^{(i^*)}, q_t^{(i^*)}, r_t^{(i^*)})\}$
13:       **if** $\text{success}(r_t^{(i^*)})$ **then**
14:         **break**
15:       **end if**
16:     **end for**
17: **end for**

---

---

**Algorithm 3** Turn-level Reward Computation in GRPO-based Retrieval

---

**Require:** Current context $\text{ctx}_t$, group size $G$, corpus $\mathcal{C}$, retriever $\mathcal{R}$, top-$k$ size $K$
1: Initialize lists: $\{\theta_{t,i}, q_{t,i}, R_{t,i}\}_{i=1}^{G}$
2: **for** $i = 1$ **to** $G$ **do**
3:     Sample think segment: $\theta_{t,i} \sim \pi_\theta(\cdot \mid \text{ctx}_t)$
4:     Sample search query: $q_{t,i} \sim \pi_\theta(\cdot \mid \text{ctx}_t, \theta_{t,i})$
5:     Retrieve documents: $\mathcal{D}_{t,i} = \mathcal{R}(q_{t,i})$
6:     Compute evaluation metrics:

$$\text{sim}_{t,i} = \max_{d \in \mathcal{D}_{t,i}} \text{sim}(q_{t,i}, d), \quad r_{t,i}^{\text{rank}} = \text{rank of document achieving maximum similarity}$$

7:     Normalize similarity and rank:

$$\sigma(\text{sim}_{t,i}) = \begin{cases} \text{sim}_{t,i}, & \text{if } \text{sim}_{t,i} \geq 0 \\ (\text{sim}_{t,i} + 1)/2, & \text{otherwise} \end{cases}, \quad \rho(r_{t,i}^{\text{rank}}) = \begin{cases} 1 - r_{t,i}^{\text{rank}}/|\mathcal{C}|, & r_{t,i}^{\text{rank}} < \infty \\ 0, & \text{otherwise} \end{cases}$$

8:     Compute reward: $R_{t,i} = 0.5 \cdot \sigma(\text{sim}_{t,i}) + 0.5 \cdot \rho(r_{t,i}^{\text{rank}})$
9: **end for**
10: Select best generation:

$$i^* = \arg\max_i R_{t,i}, \quad \theta_t^* = \theta_{t,i^*}, \quad q_t^* = q_{t,i^*}, \quad \mathcal{D}_t^* = \text{top-}k \text{ of } \mathcal{D}_{t,i^*}$$

11: Update context for next turn:

$$\text{ctx}_{t+1} = \text{ctx}_t \cup \{\theta_t^*, s_t^*, \mathcal{D}_t^*\}$$

12: **return** queries $\{q_{t,i}\}_{i=1}^{G}$, think generations $\{\theta_{t,i}\}_{i=1}^{G}$, top-$k$ documents $\mathcal{D}_t^*$, and rewards $\{R_{t,i}\}_{i=1}^{G}$

---

## C  Detailed Experimental Setup

### C.1  Model Specifications and Training Details

**Base Model Architecture**  Our Orion models are built on the LFM2 architecture, which employs a hybrid design combining 10 double-gated short-range LIV convolution blocks and 6 grouped query attention (GQA) blocks. The architecture uses a vocabulary size of 65,536 tokens with bfloat16 precision and supports context lengths up to 32,768 tokens. All models were pre-trained on approximately 10 trillion tokens with knowledge distillation from LFM1-7B as the teacher model.

**Synthetic Data Generation Models**  Our ultra-feedback pool was constructed using eight diverse language models across three families:

- **GPT Family:** GPT-4.1-mini, GPT-4o, GPT-4o-mini
- **Llama Family:** Llama 3.1-405B, Llama 3.1-70B, Llama 3.1-8B, Llama 3.2-3B
- **Qwen Family:** Qwen2.5-7B

Each model performed 5-turn iterative retrieval on the same user queries from training splits, creating diverse search behaviors. For each query $q_i$, we obtained from each model $M_j$: search traces $\{q_{i,j,t}\}_{t=1}^5$, thinking traces $\{\phi_{i,j,t}\}_{t=1}^5$, retrieval results $\{R_{i,j,t}\}_{t=1}^5$, and performance metrics $\{\cos_{i,j,t}, \mathrm{rank}_{i,j,t}\}_{t=1}^5$.

**Training Hyperparameters**  Supervised fine-tuning employed AdamW optimizer with learning rate $5 \times 10^{-5}$, weight decay 0.01, and fixed learning rate schedule. Gradient clipping was applied with maximum norm 1.0. For GRPO training, we used group size $G = 4$, KL regularization coefficient $\beta = 0.1$, and reward shifting parameter $\epsilon = 0.2$.

**Structural Token Masking**  During training, we apply differential masking to structural tokens. End tokens (`</think>`, `</search_query>`) were included as generation targets, while content within `<user_query>...</user_query>` and `<top_k_response>...</top_k_response>` spans was masked. Start tokens (`<think>`, `<search_query>`) were also masked to focus learning on reasoning content and query formulation rather than structural markers.

### C.2  Dataset Construction

**Synthetic Data Distribution**  Our 100K training corpus maintains balanced representation with each dataset contributing exactly 25%:

- MS Marco: 25K samples – web search queries
- SciFact: 25K samples – scientific claim verification
- HotpotQA: 25K samples – multi-hop reasoning
- FEVER: 25K samples – fact-checking

GRPO training used a concentrated 40K subset (10K per dataset) selected for diversity and reasoning complexity.

**Behavioral Archetype Distribution**  Our synthetic data incorporates 10 distinct search behaviors, each contributing equally (10% each), these are discussed in detail in Appendix B.2. Each archetype implements distinct exploration-exploitation strategies, from systematic coverage to failure recovery patterns.

#### C.2.1  Retrieval Environment Configuration

**Dense Retrieval Backend**  In information retrieval settings, there are two primary approaches: sparse methods like BM25 that rely on exact term matching and statistical weighting, and dense methods that encode queries and documents into continuous vector representations for semantic

Table 7: nDCG@10 scores of different search behaviors across 12 BRIGHT domains. Performance varies notably by category, with stronger results in StackExchange domains than in coding or theorem-based tasks.

| | StackExchange | | | | | | | Coding | | Theorem-based | | |
|---|---|---|---|---|---|---|---|---|---|---|---|---|
| **Algorithm** | Bio. | Earth. | Econ. | Psy. | Rob. | Stack. | Sus. | Leet. | Pony | AoPS | TheoQ. | TheoT. |
| Adaptive Context Learner | 0.209 | 0.281 | 0.189 | 0.255 | 0.139 | 0.130 | 0.155 | 0.162 | 0.045 | 0.037 | 0.074 | 0.016 |
| Random Walk Wanderer | 0.201 | 0.216 | 0.148 | 0.202 | 0.145 | 0.105 | 0.153 | 0.178 | 0.190 | 0.040 | 0.084 | 0.019 |
| Breadth-First Explorer | 0.181 | 0.214 | 0.147 | 0.178 | 0.106 | 0.077 | 0.165 | 0.093 | 0.070 | 0.046 | 0.084 | 0.005 |
| Depth-First Driller | 0.254 | 0.307 | 0.206 | 0.241 | 0.155 | 0.122 | 0.239 | 0.198 | 0.134 | 0.041 | 0.091 | 0.020 |
| Wrong-Direction Specialist | 0.225 | 0.314 | 0.166 | 0.275 | **0.162** | 0.144 | 0.213 | 0.186 | **0.234** | 0.039 | 0.103 | 0.022 |
| Early-Success Validator | 0.261 | 0.313 | **0.232** | 0.262 | 0.154 | 0.156 | 0.213 | 0.169 | 0.164 | 0.035 | **0.115** | 0.040 |
| Exploitation-Heavy Validator | **0.264** | **0.337** | 0.205 | 0.294 | 0.159 | 0.157 | 0.179 | 0.174 | 0.096 | 0.027 | 0.092 | **0.041** |
| Greedy Hill Climber | 0.238 | 0.325 | 0.205 | **0.308** | 0.155 | 0.157 | 0.203 | **0.203** | 0.104 | 0.039 | 0.079 | 0.036 |
| Best-First Hypothesis Selector | 0.244 | 0.317 | 0.206 | 0.265 | 0.146 | **0.163** | **0.244** | 0.169 | 0.129 | 0.034 | 0.090 | 0.029 |
| Multi-Beam Parallel | 0.217 | 0.266 | 0.189 | 0.253 | 0.149 | 0.156 | 0.186 | 0.112 | 0.136 | **0.060** | 0.077 | 0.031 |

Table 8: nDCG@10 performance of different search behaviours across FEVER, HotpotQA, NFCorpus, and SciFact. Bold values indicate the best-performing behaviour for each dataset.

| | Multi-Hop | | | Single-Hop |
|---|---|---|---|---|
| Algorithm | FEVER | HotpotQA | SciFact | NFCorpus |
| Early-Success Validator | 0.495 | 0.260 | 0.680 | 0.505 |
| Wrong-Direction Specialist | 0.495 | 0.537 | **0.697** | 0.543 |
| Greedy Hill Climber | 0.363 | 0.209 | 0.637 | 0.502 |
| Best-First Hypothesis Selector | 0.383 | 0.371 | 0.656 | 0.515 |
| Exploitation-Heavy Validator | 0.383 | 0.203 | 0.633 | 0.489 |
| Depth-First Driller | **0.574** | 0.566 | 0.670 | 0.538 |
| Multi-Beam Parallel | 0.347 | 0.290 | 0.634 | 0.474 |
| Adaptive Context Learner | 0.213 | 0.160 | 0.645 | 0.486 |
| Random Walk Wanderer | 0.557 | **0.651** | 0.656 | 0.536 |
| Breadth-First Explorer | 0.552 | 0.601 | 0.642 | **0.547** |

similarity matching. For all our experiments, we use dense retrieval with MiniLM-L6-v2 embeddings, which despite being a compact model (22.7M parameters) provides fast semantic search while leaving room for improvement on BEIR subsets and BRIGHT, demonstrating that learned search strategies can compensate for suboptimal retrieval backends (Thakur et al., 2021; SU et al., 2025).

## C.3 IMPLEMENTATION AND HARDWARE INFRASTRUCTURE

All experiments were conducted on NVIDIA H100 SXM GPUs with 80GB HBM3 memory using asynchronous SQL-based dense retrieval. Training used 8×H100 GPUs per node, 128 vCPUs (Intel Sapphire Rapids), and 1.6TB system memory. Models were trained with Hugging Face's `transformers` library. For inference and evaluation, we used the OpenAI API for GPT models, the Together AI API for Llama 3.1-405B, and vLLM for all others. MiniLM-L6-v2 (384-dim vectors) was used as the dense retriever across all datasets and baselines.

Table 9: Ablation results of different SFT training strategies on BRIGHT, FEVER, and HotpotQA. Metrics: nDCG@10 (N@10), Recall@100 (R@100), Mean Reciprocal Rank (MRR), and Success@10 (S@10).

| Model | BRIGHT | | | | FEVER | | | | HotpotQA | | | |
|---|---|---|---|---|---|---|---|---|---|---|---|---|
| | N@10 | R@100 | MRR | S@10 | N@10 | R@100 | MRR | S@10 | N@10 | R@100 | MRR | S@10 |
| *Base Models* | | | | | | | | | | | | |
| LFM2-1.2B | 0.104 | 0.152 | 0.089 | 0.164 | 0.435 | 0.684 | 0.377 | 0.627 | 0.545 | 0.499 | 0.507 | 0.692 |
| LFM2-700M | 0.098 | 0.141 | 0.084 | 0.153 | 0.405 | 0.632 | 0.352 | 0.580 | 0.500 | 0.451 | 0.464 | 0.636 |
| LFM2-350M | 0.062 | 0.080 | 0.053 | 0.095 | 0.284 | 0.456 | 0.243 | 0.418 | 0.376 | 0.328 | 0.345 | 0.490 |
| *SFT (Model Souping)* | | | | | | | | | | | | |
| LFM2-1.2B | **0.207** | **0.279** | **0.176** | **0.321** | **0.634** | 0.869 | **0.583** | **0.811** | **0.686** | **0.627** | **0.662** | **0.805** |
| LFM2-700M | 0.195 | 0.277 | 0.166 | 0.306 | 0.629 | **0.873** | 0.577 | 0.808 | 0.675 | 0.619 | 0.652 | 0.793 |
| LFM2-350M | 0.154 | 0.232 | 0.131 | 0.244 | 0.566 | 0.820 | 0.514 | 0.745 | 0.633 | 0.587 | 0.608 | 0.755 |
| *SFT (Curriculum Learning)* | | | | | | | | | | | | |
| LFM2-1.2B | 0.196 | 0.271 | 0.167 | 0.302 | 0.622 | 0.861 | 0.574 | 0.791 | 0.674 | 0.620 | 0.653 | 0.788 |
| LFM2-700M | 0.187 | 0.269 | 0.161 | 0.290 | 0.620 | 0.867 | 0.570 | 0.795 | 0.667 | 0.615 | 0.645 | 0.782 |
| LFM2-350M | 0.146 | 0.224 | 0.125 | 0.230 | 0.557 | 0.812 | 0.507 | 0.730 | 0.624 | 0.581 | 0.601 | 0.739 |
| *SFT (Random Shuffling)* | | | | | | | | | | | | |
| LFM2-1.2B | 0.195 | 0.271 | 0.167 | 0.302 | 0.622 | 0.861 | 0.574 | 0.791 | 0.674 | 0.620 | 0.653 | 0.788 |
| LFM2-700M | 0.187 | 0.269 | 0.161 | 0.290 | 0.620 | 0.867 | 0.570 | 0.795 | 0.667 | 0.615 | 0.645 | 0.782 |
| LFM2-350M | 0.144 | 0.222 | 0.124 | 0.226 | 0.551 | 0.806 | 0.502 | 0.720 | 0.620 | 0.578 | 0.599 | 0.733 |
| *SFT (No-Thinking)* | | | | | | | | | | | | |
| LFM2-1.2B | 0.187 | 0.275 | 0.161 | 0.292 | 0.576 | 0.842 | 0.527 | 0.750 | 0.660 | 0.619 | 0.637 | 0.782 |
| LFM2-700M | 0.180 | 0.272 | 0.152 | 0.290 | 0.557 | 0.835 | 0.506 | 0.739 | 0.655 | 0.609 | 0.630 | 0.778 |
| LFM2-350M | 0.140 | 0.221 | 0.122 | 0.222 | 0.526 | 0.820 | 0.474 | 0.713 | 0.615 | 0.585 | 0.592 | 0.738 |
| *SFT (No Special Tokens)* | | | | | | | | | | | | |
| LFM2-1.2B | 0.191 | 0.267 | 0.164 | 0.296 | 0.617 | 0.857 | 0.568 | 0.785 | 0.669 | 0.618 | 0.648 | 0.783 |
| LFM2-700M | 0.185 | 0.268 | 0.159 | 0.288 | 0.616 | 0.864 | 0.566 | 0.790 | 0.662 | 0.613 | 0.640 | 0.777 |
| LFM2-350M | 0.138 | 0.218 | 0.119 | 0.217 | 0.542 | 0.800 | 0.495 | 0.710 | 0.612 | 0.574 | 0.591 | 0.725 |

# D  ADDITIONAL ABLATIONS

## D.1  EFFECT OF STRUCTURAL MARKERS ON MULTI-TURN SEARCH

A key design question is whether explicit structural markers (`</think>`, `</search_query>`) are necessary for learning multi-turn search strategies, or whether models can develop these capabilities through implicit behavioral patterns alone.

We compare models trained with full structural scaffolding against those trained without special tokens, using random shuffling as the baseline training approach for both conditions to ensure fair comparison. As shown in Table 9 and Table 10, the results reveal surprisingly modest performance differences. On BRIGHT, removing structural tokens drops nDCG@10 from only 19.5% to 19.1% - a mere 0.4 percentage point difference. Similar minimal gaps appear across other benchmarks: FEVER (62.2% vs 61.7%) and HotpotQA (67.4% vs 66.9%).

This robustness suggests that models learn search patterns primarily from the underlying behavioral content in our synthetic data rather than relying on explicit formatting cues. The consistent alternation between reasoning and querying in our training trajectories provides sufficient implicit structure for models to internalize multi-turn search dynamics. While structural tokens improve training interpretability and debugging, they are not strictly necessary for developing adaptive search behaviors, but we consider them lightweight scaffolds. They segment reasoning (`</think>`), and querying (`</search_query>`) into clear units, making the process more interpretable and giving the model

Table 10: Ablation results of different SFT training strategies on MS Marco, NFCorpus, and Sci-Fact. Metrics: nDCG@10 (N@10), Recall@100 (R@100), Mean Reciprocal Rank (MRR), and Success@10 (S@10).

| Model | MS Marco | | | | NFCorpus | | | | SciFact | | | |
|---|---|---|---|---|---|---|---|---|---|---|---|---|
| | N@10 | R@100 | MRR | S@10 | N@10 | R@100 | MRR | S@10 | N@10 | R@100 | MRR | S@10 |
| *Base Models* | | | | | | | | | | | | |
| LFM2-1.2B | 0.727 | 0.372 | 0.758 | 0.907 | 0.538 | 0.292 | 0.506 | 0.726 | 0.630 | 0.897 | 0.576 | 0.812 |
| LFM2-700M | 0.712 | 0.333 | 0.735 | 0.915 | 0.524 | 0.289 | 0.492 | 0.708 | 0.616 | 0.865 | 0.567 | 0.787 |
| LFM2-350M | 0.506 | 0.239 | 0.499 | 0.698 | 0.481 | 0.244 | 0.448 | 0.665 | 0.559 | 0.800 | 0.502 | 0.742 |
| *SFT (Model Souping)* | | | | | | | | | | | | |
| LFM2-1.2B | 0.836 | 0.454 | 0.875 | 0.954 | **0.582** | **0.328** | **0.560** | **0.753** | **0.723** | **0.930** | **0.688** | 0.850 |
| LFM2-700M | 0.814 | 0.435 | 0.855 | 0.938 | 0.564 | 0.313 | 0.543 | 0.737 | 0.703 | 0.924 | 0.665 | 0.840 |
| LFM2-350M | 0.828 | 0.430 | 0.855 | 0.969 | 0.535 | 0.306 | 0.518 | 0.691 | 0.683 | 0.908 | 0.641 | 0.829 |
| *SFT ( Curriculum Learning)* | | | | | | | | | | | | |
| LFM2-1.2B | 0.831 | 0.453 | 0.871 | 0.946 | 0.572 | 0.322 | 0.554 | 0.734 | 0.713 | 0.922 | 0.681 | 0.835 |
| LFM2-700M | 0.814 | 0.435 | 0.855 | 0.938 | 0.554 | 0.310 | 0.536 | 0.719 | 0.697 | 0.923 | 0.662 | 0.830 |
| LFM2-350M | 0.823 | 0.426 | 0.851 | 0.961 | 0.526 | 0.301 | 0.510 | 0.676 | 0.675 | 0.901 | 0.636 | 0.815 |
| *SFT (Random Shuffling)* | | | | | | | | | | | | |
| LFM2-1.2B | 0.831 | 0.453 | 0.871 | 0.946 | 0.571 | 0.321 | 0.554 | 0.733 | 0.713 | 0.922 | 0.681 | 0.835 |
| LFM2-700M | 0.814 | 0.435 | 0.855 | 0.938 | 0.554 | 0.310 | 0.536 | 0.719 | 0.697 | 0.923 | 0.662 | 0.830 |
| LFM2-350M | 0.823 | 0.426 | 0.851 | 0.961 | 0.523 | 0.299 | 0.509 | 0.669 | 0.672 | 0.899 | 0.633 | 0.808 |
| *SFT (No-Thinking)* | | | | | | | | | | | | |
| LFM2-1.2B-FT | 0.777 | 0.431 | 0.814 | 0.922 | 0.555 | 0.325 | 0.535 | 0.724 | 0.704 | 0.927 | 0.657 | **0.862** |
| LFM2-700M-FT | **0.866** | **0.478** | **0.921** | **1.000** | 0.556 | 0.311 | 0.536 | 0.715 | 0.687 | 0.928 | 0.649 | 0.829 |
| LFM2-350M-FT | 0.846 | 0.432 | 0.887 | 0.977 | 0.515 | 0.311 | 0.491 | 0.679 | 0.659 | 0.910 | 0.617 | 0.809 |
| *SFT (No Special Tokens)* | | | | | | | | | | | | |
| LFM2-1.2B | 0.809 | 0.444 | 0.848 | 0.922 | 0.567 | 0.319 | 0.549 | 0.728 | 0.708 | 0.920 | 0.676 | 0.829 |
| LFM2-700M | 0.798 | 0.422 | 0.833 | 0.915 | 0.551 | 0.309 | 0.533 | 0.715 | 0.695 | 0.923 | 0.659 | 0.827 |
| LFM2-350M | 0.804 | 0.414 | 0.829 | 0.946 | 0.518 | 0.297 | 0.506 | 0.662 | 0.666 | 0.893 | 0.627 | 0.801 |

a simple signal of when to reflect versus retrieve. In this sense, they are less about performance gains and more about providing structure and readability.

## D.2 EFFECT OF DIFFERENT TRAINING STRATEGIES ON SFT

We analyze how different ways of incorporating behavioral archetypes influence the search strategies learned through SFT. We compare four approaches: (1) random shuffling of archetypes across training examples, (2) curriculum learning that progresses from simple reformulations to complex multi-hypothesis strategies, and (3) model souping that merges specialist models.

**Model Souping** Following SmolLM3's & Llama-Nemotron-Super's approach that uses MergeKit to combine behavioral specialists with exponential weighting favoring sophisticated strategies (Bakouch et al., 2025; Goddard et al., 2024; Bercovich et al., 2025), we combine specialist models trained on individual behavioral archetypes. The merging process uses exponential weighting where behavioral archetypes appearing later in the curriculum sequence receive higher weights in the final combination. This weighting scheme reflects the assumption that more complex search behaviors (such as multi-hypothesis coordination) are more valuable than simpler reformulation strategies. The technique allows combining the strengths of different search strategies without additional training overhead, creating a unified model that exhibits diverse search behaviors while emphasizing the most sophisticated approaches.

**Results**   Tables 9 and 10 show that model souping consistently delivers the strongest performance. On BRIGHT, the 1.2B model reaches 32.1% nDCG@10, compared to 19.6% for curriculum learning and 19.5% for random shuffling. This suggests that merging specialists preserves distinct behavioral competencies more effectively than joint training, where optimization dynamics may cause interference across archetypes. By contrast, curriculum learning provides little improvement over random shuffling. Once behavioral diversity is explicitly encoded through archetype design, temporal ordering contributes far less than diversity itself. This finding challenges the common assumption that careful pedagogical sequencing is required for complex skill acquisition, pointing instead to only behavioral diversity as the key driver.

## D.3   EFFECT OF LEARNING COMPONENTS ON SEARCH BEHAVIOR

Learning components in our framework refer to three distinct stages that each contribute to search behavior: SFT, GRPO, and inference-time beam search. We analyze how different search behaviors manifest across these training stages. To understand how different synthetic behavioral archetypes contribute to overall performance, we conducted individual algorithm studies using the ten distinct search behaviors detailed in Appendix B.2 and Table 6. These behavioral archetypes, ranging from systematic exploration strategies like breadth-first and depth-first search to adaptive refinement approaches like hill climbing and context learning, form the foundation of our SFT stage and allow us to isolate the contribution of specific search strategies across different retrieval scenarios.

The results reveal specialization patterns across task domains (Tables 7 and 8). On StackExchange domains, exploitation-heavy strategies consistently dominate, with Exploitation-Heavy Validator achieving top-3 performance in 5 of 7 BRIGHT topics and reaching 33.7% on Earth Sciences. Conversely, these same exploitation strategies perform poorly on multi-hop reasoning tasks, where exploration-based approaches like Random Walk Wanderer excel (65.1% on HotpotQA vs bottom-tier StackExchange performance). Most remarkably, Wrong-Direction Specialist shows perfect task specialization, ranking first on fact verification (SciFact: 69.7%) and coding tasks (Pony: 23.4%) while remaining mediocre on traditional Q&A. These patterns suggest that effective multi-turn search requires different behavioral strategies for different reasoning demands: systematic exploration for multi-hop reasoning, focused exploitation for domain-specific Q&A, and error-recognition capabilities for verification tasks. The clear algorithmic specialization observed across domains validates our approach of training models on diverse behavioral archetypes, as no single search strategy proves universally effective across the breadth of retrieval scenarios. This finding aligns with prior literature on domain-specific information retrieval systems that demonstrate improved performance through task-adapted search strategies (Vakkari, 1999; 2001; Sutcliffe & Ennis, 1998).

The behavioral patterns become more nuanced when examining how training stages affect search dynamics across our complete model pipeline (Figures 3 and 4). Turn-wise success distributions (Figure 3a) show the proportion of successful queries resolved at each turn, representing completion counts of successful traces divided by total successful traces. A higher Turn 1 proportion indicates that when models do succeed, they tend to succeed immediately, while more distributed patterns suggest models that can recover and succeed even after initial failures. Importantly, these distributions only reflect the composition of successful queries and do not indicate overall performance levels. Our Orion models show more distributed success patterns compared to general-purpose LLMs, indicating an ability to continue searching effectively rather than giving up after early unsuccessful attempts. Rank stagnation analysis (Figure 3b) further supports this interpretation, with our models showing substantially lower stagnation rates (2-4%) compared to general-purpose LLMs (4-8%), indicating reduced tendency toward repetitive, ineffective query patterns. Across all three model sizes, GRPO training shows reduced stagnation rates compared to SFT variants, indicating that RL may help models avoid repetitive query patterns, while the inference-time beam search approach (Main) demonstrates more varied turn-wise success distributions, showing the ability to recover.

The training stage effects become particularly evident in backtracking behavior and query efficiency metrics (Figure 4). Backtracking analysis (Figure 4a) reveals that our Main variants show the highest backtracking rates, particularly for Orion-700M and Orion-350M, indicating that inference-time beam search encourages more exploratory behavior that occasionally requires course-correction. Among the trained variants, GRPO consistently outperforms SFT across all model sizes in backtracking capability, with GRPO showing higher backtracking rates than SFT, suggesting that RL may be enabling more adaptive search strategies that can recover from suboptimal directions. How-

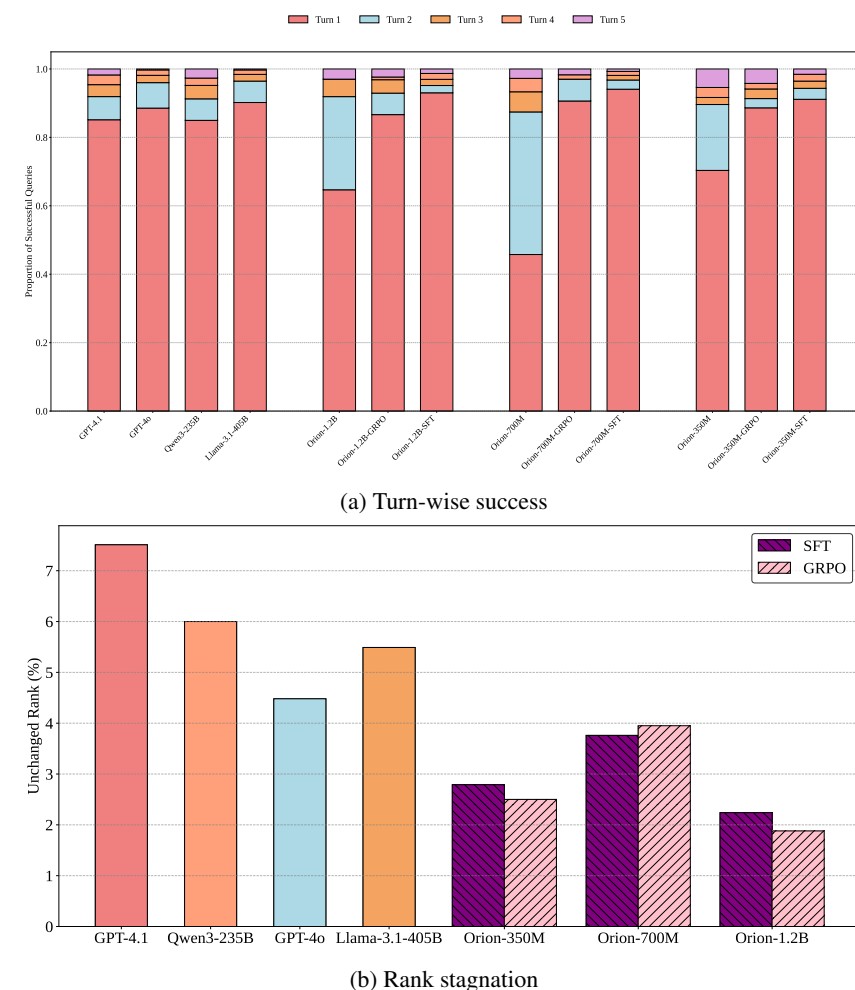

(a) Turn-wise success

(b) Rank stagnation

Figure 3: Search behavior analysis across models: (a) demonstrates how successful queries distribute across search turns for different models, while (b) illustrates the proportion of queries with unchanged rankings across turns, indicating repetitive search patterns and the inability to overcome search stagnation.

ever, the pattern is not uniform across sizes, Orion-1.2B Main shows lower backtracking than its GRPO variant. Query length distributions (Figure 4b) analyze the distribution of search queries as produced by the different models and reveal that our SFT and GRPO variants consistently generate much shorter queries, with medians substantially lower than external LLMs like Llama-3.1-405B which produces highly variable and verbose queries. The Main variants show slightly higher query lengths, potentially due to inference-time beam search encouraging more elaborate query formulations during the exploration process. Notably, the 1.2B model demonstrates the most stable behavior across training stages, with SFT, GRPO, and Main variants producing similar query length distributions, suggesting that larger model capacity leads to more consistent succinct query generation patterns regardless of the specific training approach.

### D.4    EFFECT OF DATA SCALING IN GRPO TRAINING

We evaluated the impact of training data size on GRPO by experimenting with 10K, 40K, and 80K total datapoints. Performance improved as the dataset grew, with 40K datapoints providing a strong balance between effectiveness and efficiency. While 80K datapoints yielded slightly better results, the gains were marginal relative to the increased training time and computational complexity, so we report results using 40K datapoints in the main experiments.

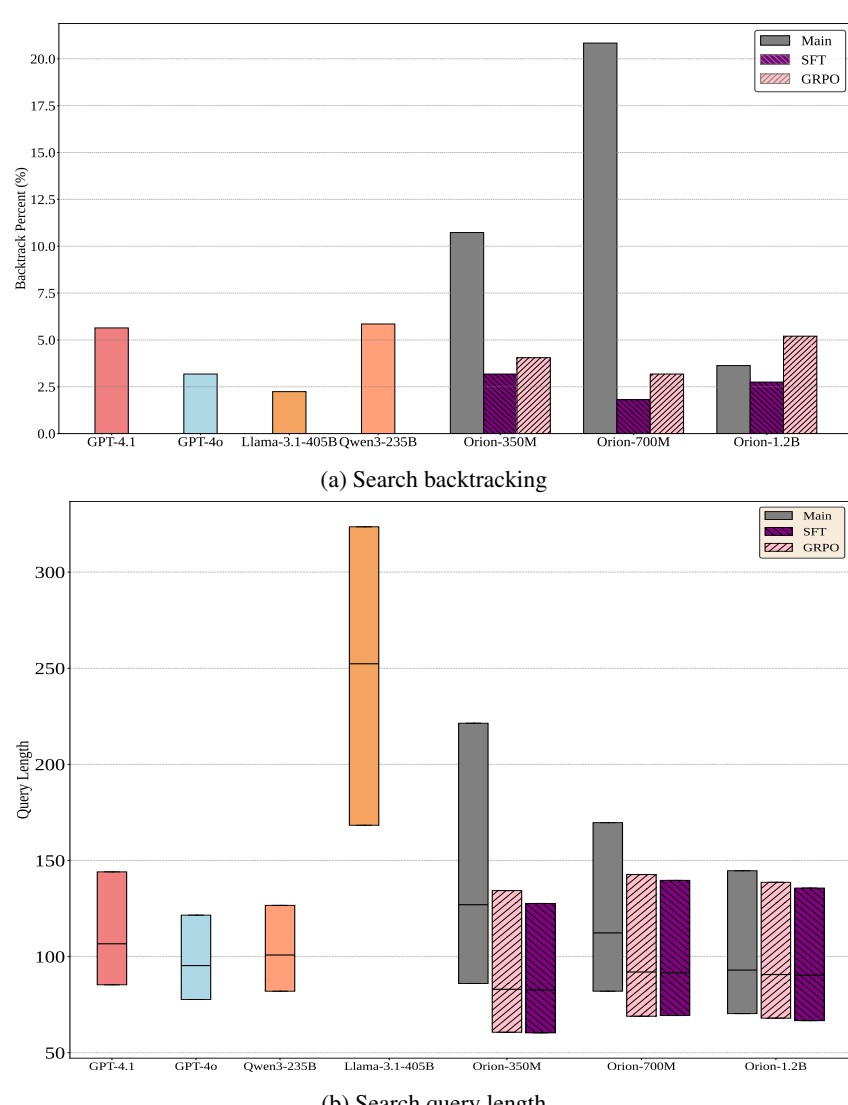

(a) Search backtracking

(b) Search query length

Figure 4: Adaptive search capabilities: (c) measures backtracking behavior by counting queries where rankings ($r$) deteriorate then recover ($r_{i-1} > r_i < r_{i+1}$), while (d) shows search query length distribution across all three Orion model variants, demonstrating our models generate relatively succinct search queries.

## D.5 EFFECT OF Z-SCORE NORMALIZATION ON REWARD COMPUTATION

We experimented with normalizing rewards via corpus-level z-scores to account for varying score distributions across the four corpora in the GRPO dataset. While this approach aimed to stabilize learning by standardizing reward magnitudes, it did not improve performance and was therefore not used in the final model.

From Table 11, we observe that adding more data slightly improves most metrics, with the largest gains seen when combined with z-score normalization. However, the improvement is marginal ($\sim$1–2%) compared to the baseline, suggesting 40K datapoints without normalization strikes the best balance between efficiency and performance.

Table 11: Ablation results on the effect of data scale in GRPO. Metrics: nDCG@10 (N@10), Recall@100 (R@100), Mean Reciprocal Rank (MRR), and Success@10 (S@10).

| Model | BRIGHT | | | | FEVER | | | | HotpotQA | | | |
|---|---|---|---|---|---|---|---|---|---|---|---|---|
| | N@10 | R@100 | MRR | S@10 | N@10 | R@100 | MRR | S@10 | N@10 | R@100 | MRR | S@10 |
| *Data Size Ablations* | | | | | | | | | | | | |
| LFM2-1.2B-10k-data | 0.208 | 0.276 | 0.175 | 0.327 | 0.631 | 0.871 | 0.579 | 0.809 | 0.685 | 0.629 | 0.661 | 0.805 |
| LFM2-1.2B-40k-data | 0.212 | 0.290 | 0.180 | 0.335 | **0.643** | 0.873 | **0.591** | 0.822 | **0.692** | 0.631 | 0.666 | 0.815 |
| LFM2-1.2B-80k-data | 0.217 | 0.285 | 0.183 | **0.342** | **0.643** | **0.879** | 0.590 | **0.823** | **0.692** | 0.631 | **0.667** | 0.813 |
| *Z-Score Normalization Ablations* | | | | | | | | | | | | |
| LFM2-1.2B-10k-data-z-score | 0.211 | 0.279 | 0.178 | 0.334 | 0.636 | 0.872 | 0.583 | 0.816 | 0.691 | 0.630 | 0.666 | 0.813 |
| LFM2-1.2B-40k-data-z-score | **0.216** | **0.292** | **0.186** | 0.329 | 0.642 | 0.873 | 0.589 | 0.821 | **0.692** | **0.632** | **0.667** | **0.817** |

Table 12: Ablation results on the effect of z-score normalization in GRPO reward computation. Metrics: nDCG@10 (N@10), Recall@100 (R@100), Mean Reciprocal Rank (MRR), and Success@10 (S@10).

| Model | MS Marco | | | | NFCorpus | | | | SciFact | | | |
|---|---|---|---|---|---|---|---|---|---|---|---|---|
| | N@10 | R@100 | MRR | S@10 | N@10 | R@100 | MRR | S@10 | N@10 | R@100 | MRR | S@10 |
| *Data Size Ablations* | | | | | | | | | | | | |
| LFM2-1.2B-10k-data | 0.845 | 0.436 | 0.884 | 0.969 | 0.591 | 0.321 | **0.571** | 0.757 | 0.711 | 0.929 | 0.673 | 0.848 |
| LFM2-1.2B-40k-data | 0.842 | **0.478** | 0.881 | 0.992 | 0.583 | **0.328** | 0.554 | 0.765 | **0.732** | 0.925 | **0.693** | **0.870** |
| LFM2-1.2B-80k-data | 0.840 | 0.459 | 0.889 | 0.953 | **0.594** | 0.324 | 0.567 | **0.769** | 0.707 | 0.928 | 0.669 | 0.842 |
| *Z-Score Normalization Ablations* | | | | | | | | | | | | |
| LFM2-1.2B-10k-data-z-score | 0.867 | 0.462 | **0.939** | 0.969 | 0.585 | 0.324 | 0.566 | 0.748 | 0.720 | 0.928 | 0.685 | 0.848 |
| LFM2-1.2B-40k-data-z-score | **0.874** | 0.465 | 0.913 | **1.000** | 0.578 | 0.321 | 0.552 | 0.759 | 0.719 | **0.934** | 0.675 | **0.870** |

# E  COMPLETE SET OF EVALUATION RESULTS

Tables 13 and 14 report the full evaluation results across all datasets, including additional metrics beyond the main text. For each metric, we show the "+/-" relative to a strong baseline (e.g., GPT-4.1 or Llama 3.1-405B), chosen per dataset to provide a clear and interpretable measure of Orion's improvements over established models.

Table 13: Complete evaluation results for BRIGHT, FEVER, and HotpotQA. Metrics: Success@10 (S@10), nDCG@10 (N@10), Recall@100 (R@100), and Mean Reciprocal Rank (MRR). *Parameter counts approximated by Abacha et al. 2025; †values not evaluated due to unavailable model checkpoints.

| | BRIGHT (macro average) | | | | FEVER | | | | HotpotQA | | | |
|---|---|---|---|---|---|---|---|---|---|---|---|---|
| Model | S@10 | nDCG@10 | R@100 | MRR | S@10 | nDCG@10 | R@100 | MRR | S@10 | nDCG@10 | R@100 | MRR |
| *General Purpose LLMs* | | | | | | | | | | | | |
| GPT-4.1 | 0.351 | 0.222 | 0.323 | 0.187 | 0.826 | 0.613 | 0.869 | 0.548 | **0.898** | **0.748** | **0.696** | **0.719** |
| GPT-4.1-mini | 0.335 | 0.213 | 0.320 | 0.180 | 0.794 | 0.588 | 0.851 | 0.527 | 0.867 | 0.713 | 0.663 | 0.681 |
| GPT-4o (200B*) | 0.307 | 0.191 | 0.294 | 0.163 | 0.803 | 0.595 | 0.854 | 0.533 | 0.889 | 0.736 | 0.678 | 0.704 |
| GPT-4o-mini (8B*) | 0.276 | 0.172 | 0.278 | 0.147 | 0.752 | 0.549 | 0.828 | 0.490 | 0.842 | 0.686 | 0.646 | 0.651 |
| Llama 3.1-405B | 0.304 | 0.192 | 0.311 | 0.162 | 0.847 | 0.648 | 0.890 | 0.588 | 0.896 | 0.738 | 0.692 | 0.706 |
| Llama 3.1-70B | 0.295 | 0.186 | 0.306 | 0.159 | 0.840 | 0.634 | 0.885 | 0.573 | 0.886 | 0.727 | 0.678 | 0.693 |
| Llama 3.1-8B | 0.277 | 0.172 | 0.278 | 0.146 | 0.808 | 0.597 | 0.860 | 0.533 | 0.840 | 0.681 | 0.642 | 0.646 |
| Llama 3.2-3B | 0.228 | 0.141 | 0.241 | 0.123 | 0.788 | 0.576 | 0.850 | 0.512 | 0.818 | 0.666 | 0.621 | 0.632 |
| Qwen3-235B | 0.345 | 0.218 | 0.325 | 0.184 | 0.809 | 0.602 | 0.859 | 0.541 | 0.881 | 0.725 | 0.665 | 0.691 |
| Qwen2.5-7B | 0.268 | 0.165 | 0.262 | 0.139 | 0.779 | 0.567 | 0.849 | 0.504 | 0.818 | 0.668 | 0.623 | 0.635 |
| Qwen2.5-3B | 0.222 | 0.135 | 0.251 | 0.117 | 0.751 | 0.547 | 0.834 | 0.488 | 0.793 | 0.648 | 0.614 | 0.617 |
| *Retrieval Baselines* | | | | | | | | | | | | |
| BM25 (dense) | 0.272 | 0.097 | 0.315 | 0.127 | 0.827 | 0.578 | 0.935 | 0.540 | 0.954 | 0.619 | 0.827 | 0.802 |
| BM25 (sparse) | 0.194 | 0.070 | 0.225 | 0.091 | 0.689 | 0.482 | 0.850 | 0.450 | 0.867 | 0.563 | 0.752 | 0.729 |
| DeepRetrieval (3B)† | N/A | N/A | N/A | N/A | N/A | N/A | N/A | N/A | N/A | N/A | N/A | N/A |
| *Orion Models (ours)* | | | | | | | | | | | | |
| Orion-Large | 0.375 | 0.249 | 0.338 | 0.210 | 0.859 | **0.653** | 0.892 | **0.598** | 0.848 | 0.716 | 0.652 | 0.690 |
| | (+0.024) | (+0.027) | (+0.015) | (+0.023) | (+0.012) | (+0.005) | (+0.002) | (+0.010) | (-0.050) | (-0.032) | (-0.044) | (-0.029) |
| Orion-Medium | **0.426** | **0.257** | **0.406** | 0.209 | **0.862** | 0.633 | **0.906** | 0.573 | 0.836 | 0.685 | 0.634 | 0.659 |
| | (+0.075) | (+0.035) | (+0.083) | (+0.022) | (+0.015) | (-0.015) | (+0.016) | (-0.015) | (-0.062) | (-0.063) | (-0.062) | (-0.060) |
| Orion-Small | 0.338 | 0.241 | 0.378 | 0.201 | 0.804 | 0.577 | 0.865 | 0.518 | 0.809 | 0.641 | 0.611 | 0.615 |
| | (-0.013) | (+0.019) | (+0.055) | (+0.014) | (-0.043) | (-0.071) | (-0.025) | (-0.070) | (-0.089) | (-0.107) | (-0.085) | (-0.104) |

Table 14: Complete evaluation results for MS Marco, NFCorpus, and SciFact. Metrics: Success@10 (S@10), nDCG@10 (N@10), Recall@100 (R@100), and Mean Reciprocal Rank (MRR). *Parameter counts approximated by Abacha et al. 2025; †values not evaluated due to unavailable model checkpoints.

| | MS Marco | | | | NFCorpus | | | | SciFact | | | |
|---|---|---|---|---|---|---|---|---|---|---|---|---|
| Model | S@10 | nDCG@10 | R@100 | MRR | S@10 | nDCG@10 | R@100 | MRR | S@10 | nDCG@10 | R@100 | MRR |
| *General Purpose LLMs* | | | | | | | | | | | | |
| GPT-4.1 | 0.992 | 0.877 | 0.463 | 0.944 | 0.753 | 0.578 | 0.322 | 0.547 | 0.876 | 0.724 | 0.950 | 0.680 |
| GPT-4.1-mini | 0.992 | 0.857 | 0.470 | 0.890 | 0.738 | 0.565 | 0.332 | 0.536 | 0.886 | 0.723 | 0.950 | 0.675 |
| GPT-4o (200B*) | **1.000** | 0.861 | 0.465 | 0.888 | 0.736 | 0.558 | 0.326 | 0.533 | 0.880 | 0.708 | 0.937 | 0.658 |
| GPT-4o-mini (8B*) | 0.992 | 0.844 | 0.439 | 0.883 | 0.728 | 0.537 | 0.314 | 0.503 | 0.858 | 0.698 | 0.941 | 0.654 |
| Llama 3.1-405B | 0.992 | 0.849 | 0.469 | 0.897 | 0.722 | 0.562 | 0.333 | 0.547 | 0.869 | 0.703 | **0.957** | 0.656 |
| Llama 3.1-70B | 0.977 | 0.839 | 0.464 | 0.873 | 0.726 | 0.565 | 0.339 | 0.545 | 0.875 | 0.707 | 0.946 | 0.658 |
| Llama 3.1-8B | 0.992 | 0.838 | 0.432 | 0.869 | 0.740 | 0.553 | 0.324 | 0.523 | 0.871 | 0.702 | 0.952 | 0.653 |
| Llama 3.2-3B | 0.977 | 0.846 | 0.436 | 0.904 | 0.739 | 0.550 | 0.313 | 0.520 | 0.831 | 0.671 | 0.933 | 0.624 |
| Qwen3-235B | **1.000** | 0.870 | 0.477 | 0.940 | 0.745 | 0.570 | 0.327 | 0.543 | 0.884 | 0.726 | 0.950 | 0.679 |
| Qwen2.5-7B | 0.992 | 0.848 | 0.459 | 0.902 | 0.733 | 0.551 | 0.321 | 0.524 | 0.863 | 0.692 | 0.936 | 0.643 |
| Qwen2.5-3B | 0.962 | 0.823 | 0.445 | 0.875 | 0.737 | 0.563 | 0.322 | 0.545 | 0.847 | 0.672 | 0.930 | 0.621 |
| *Retrieval Baselines* | | | | | | | | | | | | |
| BM25 (dense) | 0.945 | 0.835 | 0.537 | 0.884 | 0.769 | 0.374 | 0.264 | 0.575 | 0.808 | 0.661 | 0.888 | 0.614 |
| BM25 (sparse) | 0.756 | 0.642 | 0.398 | 0.691 | 0.641 | 0.267 | 0.211 | 0.471 | 0.703 | 0.560 | 0.793 | 0.529 |
| DeepRetrieval (3B)† | N/A | N/A | N/A | N/A | N/A | N/A | N/A | N/A | N/A | N/A | N/A | N/A |
| *Orion Models (Ours)* | | | | | | | | | | | | |
| Orion-Large | 0.992 | 0.849 | **0.482** | 0.887 | 0.804 | **0.632** | **0.338** | **0.603** | 0.882 | **0.776** | **0.957** | **0.735** |
| | (-0.008) | (-0.021) | (+0.005) | (-0.053) | (+0.051) | (+0.054) | (+0.016) | (+0.056) | (+0.006) | (+0.052) | (+0.007) | (+0.055) |
| Orion-Medium | 0.984 | **0.890** | 0.443 | 0.926 | **0.809** | 0.605 | 0.336 | 0.573 | 0.854 | 0.711 | 0.934 | 0.665 |
| | (-0.016) | (+0.020) | (-0.034) | (+0.014) | (+0.056) | (+0.027) | (+0.014) | (+0.026) | (-0.022) | (-0.013) | (-0.016) | (-0.015) |
| Orion-Small | 0.977 | 0.874 | 0.476 | **0.946** | 0.782 | 0.577 | 0.323 | 0.545 | 0.866 | 0.709 | 0.932 | 0.671 |
| | (-0.023) | (+0.004) | (-0.001) | (+0.006) | (+0.029) | (-0.001) | (+0.001) | (-0.002) | (-0.010) | (-0.015) | (-0.018) | (-0.009) |

# F  ORION EXAMPLE

Table 15: Case study on HotpotQA: Orion successfully backtracks and recovers the target document, while GPT-4o loops without recovery. Considering top-5 as success.

---

**Question:** In which professional wrestling promotion did Randy Savage perform, after he left the World Wrestling Federation?

**Ground Truth:** World Championship Wrestling (WCW)

---

**Orion-1.2B (successful)**

```
Rank progression:  12 → 13 → 8 → 1 (success)
Backtracking:  Yes
```

<think> As I embark on this search, I need to initialize my beam search strategy to explore the various paths related to Randy Savage's career after leaving the World Wrestling Federation. My first step is to establish multiple hypotheses about where he might have transitioned his career. I can envision several beams: one focusing on his post-FW career, another on his professional wrestling promotions, and a third on his overall career trajectory. I will start by formulating a beam that directly addresses the question of his post-FW career. With this structured approach, I can formulate my search query: "Randy Savage professional wrestling promotion after leaving World Wrestling Federation." This query encapsulates my intent to explore the connections between his career changes and the specific promotions he was involved with after leaving the WWE."</think>

<search> Randy Savage professional wrestling promotion after leaving World Wrestling Federation </search>

<top-k> Doc mentions Randy Savage in WCW. </top-k>

<think> As I delve into the next phase of my search, I need to acknowledge the parallel beam that focuses on "Randy Savage professional wrestling promotion after leaving World Wrestling Federation."This beam presents a promising avenue, as it directly addresses the user's query about the specific promotion where Randy Savage left the WWF. However, I must also consider the broader context of his career trajectory and the various promotions he may have been involved with.</think>

<search> Randy Savage WCW career </search>

⇒ **Final Rank: 1, success.**

---

**GPT-4o (failed)**

```
Rank progression:  7 → 11 → 11 → 8 → 7 (failure)
Backtracking:  No
```

<think> I will identify key terms related to Randy Savage's career post-World Wrestling Federation, focusing on wrestling promotions active during that period. Then, I will search for documents containing these terms, emphasizing semantic similarity to capture relevant information. </think>

<search> Randy Savage wrestling promotion after World Wrestling Federation </search>

<top-k> Docs mention WWF history but not WCW. </top-k>

<think> The search results did not directly address Randy Savage's career immediately after leaving the World Wrestling Federation. I will refine the search query to focus specifically on Randy Savage's next wrestling promotion, emphasizing his transition timeline and the major promotions he joined. </think>

<search> Randy Savage wrestling promotion after WWF transition timeline </search>

⇒ **Final Rank: 7, target not found.**

---

# G  PROMPTS

---

**Baselines: Planning Phase (Turn $N$)**

This is an information retrieval task. Your goal is to find documents that are relevant to this target query: "{original_query}"

Turn 1 Analysis: {turn_1_think_resp}
Turn 1 Search Query: {turn_1_search_q}
Top-5 results:
{turn_1_results_text}

Turn 2 Analysis: {turn_2_think_resp}
Turn 2 Search Query: {turn_2_search_q}
Top-5 results:
{turn_2_results_text}

.
.
.

Turn {n − 1} Analysis: {turn_n − 1_think_resp}
Turn {n − 1} Search Query: {turn_n − 1_search_q}
Top-5 results:
{turn_n − 1_results_text}

Analyze the search results from your previous query. Write exactly 2 sentences (under 40 words total) explaining what happened and how you plan on improving the search query to better retrieve the target document based on the user query.

---

**Baselines: Search Query Phase (Turn $N$)**

This is an information retrieval task. Your goal is to find documents that are relevant to this target query: "{original_query}"

Turn 1 Analysis: {turn_1_think_resp}
Turn 1 Search Query: {turn_1_search_q}
Top-5 results:
{turn_1_results_text}

Turn 2 Analysis: {turn_2_think_resp}
Turn 2 Search Query: {turn_2_search_q}
Top-5 results:
{turn_2_results_text}

.
.
.

Turn {n − 1} Analysis: {turn_n − 1_think_resp}
Turn {n − 1} Search Query: {turn_n − 1_search_q}
Top-5 results:
{turn_n − 1_results_text}

Turn {n} Analysis: {turn_n_planning_response} Based on your analysis above, generate a new search query to find the target documents. Output ONLY the search query text. No

---

explanations, no quotes, no formatting, no XML tags, no JSON - just plain text for semantic similarity search.

**Orion: Complete Operation (Turn $N$)**

```
<user_query>{original_query}</user_query>

<think>{turn_1_think_response}</think>

<search_query>{turn_1_search_query}</search_query>

<top_k_response>
1. {turn_1_result_1_text}
2. {turn_1_result_2_text}
3. {turn_1_result_3_text}
4. {turn_1_result_4_text}
5. {turn_1_result_5_text}
</top_k_response>

<think>{turn_2_think_response}</think>

<search_query>{turn_2_search_query}</search_query>

<top_k_response>
1. {turn_2_result_1_text}
2. {turn_2_result_2_text}
3. {turn_2_result_3_text}
4. {turn_2_result_4_text}
5. {turn_2_result_5_text}
</top_k_response>

.
.
.

<think>{turn_n − 1_think_response}</think>

<search_query>{turn_n − 1_search_query}</search_query>

<top_k_response>
1. {turn_n − 1_result_1_text}
2. {turn_n − 1_result_2_text}
3. {turn_n − 1_result_3_text}
4. {turn_n − 1_result_4_text}
5. {turn_n − 1_result_5_text}
</top_k_response>

<think>{turn_n_think_response}</think>

<search_query>
```

