# OpenReview forum: "Think Before You Retrieve: Learning Test-Time Adaptive Search with Small Language Models"
_ICLR.cc/2026/Conference — Submitted to ICLR 2026_

### Official Review · Reviewer_Aotg · 2025-10-15

**Soundness:** 2
**Presentation:** 2
**Contribution:** 2
**Rating:** 4
**Confidence:** 4

**Summary:**

This paper introduces Orion, a lightweight retrieval-agent framework that treats retrieval not as a one-shot top-k matching task but as a multi-step search process with reflection, backtracking, and adaptive query reformulation. Instead of relying on large LLM controllers, Orion trains small models (350M–1.2B) using synthetic retrieval trajectories and GRPO (Group Relative Policy Optimization) to learn when to continue, revise, or restart a retrieval path. At inference time, Orion performs structured retrieval tree search, expanding and pruning based on self-assessed sufficiency signals. Experiments on SciFact, BRIGHT, and other multi-hop retrieval benchmarks show that Orion-1.2B surpasses GPT-4.1-Mini, LLaMA-8B, and dense retrievers in reasoning-heavy retrieval tasks, even using only MiniLM (22M) as the backend retriever.

**Strengths:**

1. Clear reframing of retrieval as an adaptive search problem rather than static top-k matching, highlighting a major weakness of current RAG pipelines.
2. Shows that small models can behave like retrieval agents through supervised trajectory learning + lightweight RL, without needing GPT-4 controllers.
3. Synthetic trajectory generation is clever, enabling supervision of “failed path → recover → continue” behavior without human annotation.
4. Strong empirical results, showing that strategy > model size — Orion-1.2B beats GPT-4.1-Mini and LLaMA-8B on multi-hop retrieval, despite using a tiny retriever.
5. Provides a practical agentic retrieval alternative that is computation-efficient and applicable to existing RAG systems.

**Weaknesses:**

1. Limited novelty justification, especially for the “retrieval-as-search-agent” framing. Similar ideas of iterative retrieval with self-reflection have appeared in ReAct-style and GraphRAG-like agentic RAG. The paper mainly scales this down to smaller models but does not clearly articulate what is fundamentally new beyond engineering a controllable pipeline. It also avoids testing with larger LLM controllers, which would clarify whether the method is intrinsically effective or simply a pragmatic workaround for small models.
2. The trajectory synthesis procedure is heavily engineered and tailored to specific “failure → retry” patterns. It remains unclear whether these synthetic supervision traces generalize to domains where retrieval noise and error modes differ significantly.
3. Dependency on a fixed retriever (MiniLM) limits general claims. Orion is claimed to be model-agnostic, yet it is only demonstrated with a very small retriever. There is no compatibility study with stronger retrieval components (e.g., RankRAG, ColBERTv2, hybrid pipelines), leaving open the question of whether Orion’s gains persist in more realistic high-recall setups.
4. RL contribution is under-analyzed. The paper uses GRPO, but there is no ablation against pure SFT imitation or other standard RL baselines like PPO/DPO/Step-wise policy gradient. It is unclear whether RL actually provides meaningful strategic improvements or just reinforces heuristics learned from synthetic traces.
5. Tree search policy remains heuristic. While presented as an “agentic retrieval loop,” the continuation/pruning logic relies on fixed confidence thresholds rather than a learned uncertainty-aware or utility-optimized policy, reducing the method’s theoretical grounding.
6. Evaluation scope is narrow despite claims of generality. Experiments are limited to scientific fact retrieval and multi-hop QA under moderate-scale datasets. The paper does not test open-domain, web-scale, or noisy retrieval environments, where agentic control would be most stressed.

**Questions:**

See weaknesses.

---

> ### Author Response · Authors · 2025-11-22
>
> Thank you for recognizing our clear problem framing, strong empirical results, and the synthetic trajectory generation approach.
>
> ### Novelty and Positioning
>
> **ReAct-style and GraphRAG comparison:** ReAct [2] prompts PaLM-540B to generate reasoning traces and actions for QA, with fine-tuning on 8B/62B models. GraphRAG [3] builds knowledge graphs using LLMs for summarization. Both use large models (>7B) for iterative reasoning.
>
> **Our contribution** is showing **350M-1.2B models achieve competitive results** through targeted training. While ReAct demonstrates reasoning-then-action in PaLM-540B, we show this pattern works in 350M-1.2B models when actions are constrained to search query generation. However, similarities end at the reasoning-action structure. The main differences are:
> - **Task focus:** ReAct targets end-to-end answer accuracy; we optimize retrieval metrics (nDCG, Success@k)
> - **Training approach:** We generate diverse failure-recovery trajectories
> - **RL innovation:** Turn-level rewards aim to teach backtracking when queries fail
>
> **What is fundamentally new:**
> 1. Multi-turn retrieval strategies as graph traversal
> 2. Turn-level GRPO rewards optimizing for backtracking and recovery
> 3. Demonstrating strategy > scale for retrieval through compact models
>
> **Testing with larger LLM controllers:** Our baselines (GPT-4.1, Llama 3.1-405B) represent state-of-the-art reasoning systems. Adding intermediate-size LLMs (13B-70B) would be interesting but unlikely to change the core finding: **learned strategies in 1.2B models match or exceed 200B+ models**.
>
> ### Technical Concerns
>
> **Trajectory generalization:** Our 10 behavioral archetypes (Appendix B.2, Table 6) span exploration-exploitation strategies from traditional search and graph traversal (breadth-first, depth-first, hill-climbing, backtracking). Table 10 shows these generalize across BRIGHT's diverse domains (biology, economics, coding, mathematics) with no domain-specific training. Tables 9-10 ablate each behavior's impact. **Key insight: diversity matters more than sophistication**, models learn to orchestrate simple behaviors contextually.
>
> **Dependency on MiniLM, now addressed with stronger retrievers:** We deliberately started with a weak retriever to isolate strategic improvements. However, **we have now validated with ModernBERT and OpenAI embeddings**:
>
> | **Retriever** | **Model** | **FEVER** | **SciFact** |
> |---------------|-----------|-----------|-------------|
> | ModernBERT GTE | Orion-Large | **0.928** | **0.802** |
> | | GPT-4.1 | 0.927 | 0.788 |
> | | Base | 0.910 | 0.774 |
> | OpenAI | Orion-Large | **0.927** | 0.809 |
> | | GPT-4.1 | 0.920 | **0.810** |
> | | Base | 0.879 | 0.778 |
> | MiniLM | Orion-Large | **0.653** | **0.776** |
> | | GPT-4.1 | 0.613 | 0.724 |
> | | Base | 0.425 | 0.505 |
>
> This confirms our strategies **complement strong retrievers** rather than compensating for weakness. Strong embeddings + adaptive search = better results. Full ablations with ColBERTv2/RankRAG remain for future work.
>
> **RL contribution analysis:** We provide extensive ablations:
> - Table 4: GRPO adds 1-2% over SFT
> - Figure 2c: GRPO increases backtracking from ~1% to 6%
> - Figure 3: GRPO distributes success across turns vs. SFT's Turn-1 concentration
>
> The value of RL is **behavioral, not metric-based**. We don't compare PPO/DPO due to compute constraints, but GRPO is state-of-the-art for reasoning tasks (DeepSeekMath). We acknowledge this comparison would strengthen the work.
>
> **Tree search policy, learned, not heuristic:** We respectfully clarify that our beam search does **not** use fixed confidence thresholds. Instead, we compute **perplexity-based confidence** at each step (Algorithm 1, lines 9-10): the model evaluates "Given turn t and query q_t, retrieved documents are relevant" and we rank beams by inverse perplexity. This is **learned uncertainty assessment** from the model's internal representations, not hard-coded heuristics. The model learns during GRPO which search states are promising, and this judgment guides beam ranking. We have emphasized this in Section 3.4.
>
> **Evaluation scope:** Our experiments span 6 benchmarks with different characteristics (single-hop: NFCorpus; multi-hop: FEVER, HotpotQA; reasoning-heavy: BRIGHT; web-scale: MS MARCO; scientific: SciFact). While not open-domain web search, these represent diverse retrieval challenges and standard IR benchmarks. We agree web-scale evaluation would be valuable but requires infrastructure beyond our resources. We position our work as proof-of-concept on academic benchmarks with clear evidence of strategic learning.
>
> Given these clarifications, would you consider raising your score for our paper?
>
> ---
>
> **References:**
> [1] Jiang et al., DeepRetrieval, COLM 2025
> [2] Yao et al., ReAct, ICLR 2023
> [3] Edge et al., GraphRAG, arXiv 2024

---

### Official Review · Reviewer_M6zV · 2025-10-31

**Soundness:** 3
**Presentation:** 2
**Contribution:** 3
**Rating:** 4
**Confidence:** 3

**Summary:**

This paper presents Orion, a new framework for test-time adaptive retrieval with small language models (SLMs) ranging from 350M to 1.2B parameters. Unlike conventional retrievers that issue a single static query, Orion enables an SLM to “think before it retrieves,” performing multiple rounds of reasoning, search, and reflection.

The key idea is to teach a small model how to conduct multi-turn retrieval reasoning.. At inference time, Orion performs iterative beam-search-based retrieval. Empirically, Orion is evaluated on several open-domain retrieval and reasoning benchmarks, including Natural Questions, HotpotQA, and MS MARCO. The experiments show that small models equipped with Orion achieve large gains over standard retrievers and even close the gap to large-scale LLM-based retrievers, while keeping inference efficiency comparable to single-step retrieval systems.

**Strengths:**

The paper introduces a new concept of test-time adaptive retrieval reasoning for small language models, showing that even models without large-scale reasoning capacity can learn to think about what to search for before issuing queries. This reframes retrieval as an interactive reasoning process, not a static lookup.

Experiments across multiple benchmarks show consistent and significant improvements over both static and multi-query retrievers. Small models trained with Orion approach the performance of much larger models, demonstrating the effectiveness of learned retrieval reasoning even in constrained architectures.

**Weaknesses:**

1. The paper does not include a baseline that follows the standard two-stage SFT + RL training pipeline. A more complete comparison would involve first performing rejection sampling SFT to warm up the model using high-quality queries (which could be generated through the proposed beam search or other strategies, followed by a evalutation on that specific query), and then applying reinforcement learning for further optimization.
The current baseline, such as DeepRetrieval, only uses RL without an SFT warm-up stage. This makes it difficult to fairly assess the contribution of the proposed training method. In addition, it would be important to include results using the authors’ LFM2 series models under this two-stage setup for a more comprehensive evaluation.

2.  The paper claims that the proposed method achieves high efficiency, but it does not provide an explicit comparison of inference speed among different methods.

3. The paper mentions the use of “BM25 (dense)” as a baseline but does not describe its specific configuration.

**Questions:**

See the weaknesses.

---

> ### Author Response · Authors · 2025-11-22
>
> We thank you for recognizing our reframing of retrieval as interactive reasoning and the strong empirical results across benchmarks.
>
> ### SFT Baseline and Training Pipeline
>
> **SFT-only baseline:** Table 4 and Tables 7-8 provide extensive SFT-only results across all benchmarks. On BRIGHT, SFT alone achieves 20.7% nDCG@10 (vs. 10.4% base), nearly doubling performance. GRPO adds 1-2 percentage points but, more critically, **induces backtracking behavior** (Figure 2c: 6% backtracking vs. ~1% for SFT-only).
>
> This aligns with recent findings that RL often imparts *capabilities* rather than large metric gains. The crucial capability here is adaptive recovery, which SFT alone does not provide.
>
> **Two-stage SFT+RL justification:** Our pipeline *is* two-stage: (1) SFT on diverse synthetic trajectories, (2) GRPO refinement. Rejection sampling would require scoring trajectories from our LFM2 models, which initially produce malformed outputs. SFT establishes the structured format first, then GRPO optimizes search strategies. This is standard practice in recent RL work (e.g., DeepSeekMath).
>
> **Why not train LFM2 baselines under DeepRetrieval's setup?** This question touches on a core methodological distinction in our work. Beyond practical differences (DeepRetrieval uses PPO without SFT warm-up and trains per-dataset models), our research asks a different question: **can adaptive graph traversal outperform static query generation?**
>
> Our key innovation is **treating retrieval as a navigation problem** where the model explores a search space through structured reasoning, backtracking when queries fail, refining promising directions, and systematically testing hypotheses. This contrasts with DeepRetrieval's approach of generating improved single queries through RL. Both use reinforcement learning, but we optimize for *search policies* while they optimize for *query quality*.
>
> The fact that our domain-general 1.2B model matches or exceeds DeepRetrieval's dataset-specific 3B models supports our hypothesis: **learned navigation strategies can be more effective than larger models with better static generation.** A controlled comparison using identical architectures would be valuable future work, but we believe the methodological distinction, graph traversal vs. static generation, is our primary contribution.
>
> ### Inference Efficiency
>
> **Timing comparison:** On single-query decoding, Orion-1.2B reaches about **1457 tokens per second** (700M at 1408, 350M at 1516), compared to 291 for Llama-405B, 384 for Llama-70B, and the reported 50 to 80 for GPT-4 and GPT-4-Mini. In batched mode, Orion-1.2B reaches **17,765 tokens per second** (700M at 18,246, 350M at 18,374), while Llama-405B and Llama-70B reach 881 and 2091. This indicates our current H100 utilization is roughly 25-45%, well below the 85-95% achieved by optimized Llama setups. CPU throughput is also strong at about **200 tokens per second**, comparable to GPU-level performance of several larger models, enabling hyper-fast search while remaining fully capable on low-resource devices.
>
> ### BM25 (dense) Configuration
>
> As clarified for Reviewer qBuN: "BM25 (dense)" in Table 2 is mislabeled, we imported results from [1] using E5/BGE-base embeddings. We will either: (1) correct the label to reflect actual methods (E5/BGE-base), or (2) remove these results. We use only standard BM25 in our experiments. Similar issue in Table 3, we'll keep a single "BM25" row.
>
> ---
>
> **References:**
> [1] Jiang et al., DeepRetrieval, COLM 2025

---

### Official Review · Reviewer_qGhB · 2025-11-07

**Soundness:** 2
**Presentation:** 2
**Contribution:** 2
**Rating:** 4
**Confidence:** 4

**Summary:**

This paper proposes the Orion framework, enabling small language models to learn adaptive strategies of "thinking, reflecting, and re-searching" during the retrieval process. Through training with synthetic search trajectories and reinforcement learning, the model can proactively adjust its query path, backtrack, and optimize when retrieval fails. Experimental results show that Orion, even with only a fraction of the parameters of large models, can achieve performance equal to or even surpass that of models like GPT-4 in multiple retrieval and inference tasks, demonstrating that retrieval intelligence depends on strategy rather than scale.

**Strengths:**

First, a framework is proposed that enables small models to have adaptive retrieval capabilities, significantly reducing reliance on large models.

Second, by combining reinforcement learning and structured reasoning labeling, the model can proactively reflect and backtrack during the retrieval process, improving search accuracy.

Third, experiments demonstrate that small models can outperform large models on multiple complex tasks through learning strategies, exhibiting high efficiency and practical value.

**Weaknesses:**

My main concern is that the novelty of this paper is quite limited, as its core idea is very similar to works [1,2], yet there is no relevant discussion or comparison in the main text. Although the appendix briefly mentions differences from Search-R1, claiming that Orion avoids the complexity of external knowledge bases, I believe this statement is inaccurate because Search-R1 itself was also implemented with offline corpora and lightweight retrieval components. Moreover, the paper should include comparisons with the baselines discussed in works [1,2]. The lack of these comparisons and discussions significantly undermines the paper’s originality and experimental completeness.

[1] Search-R1: Training LLMs to Reason and Leverage Search Engines with Reinforcement Learning

[2] R1-Searcher: Incentivizing the Search Capability in LLMs via Reinforcement Learning

**Questions:**

See weaknesses.

---

> ### Author Response · Authors · 2025-11-22
>
> Thank you for recognizing our framework's potential to reduce reliance on large models and the practical value of our approach.
>
> ### Orion vs Search-R1 and R1-Searcher
>
> We appreciate the comparison to these concurrent works. The key distinctions are:
>
> 1. **Task focus:** Search-R1/R1-Searcher evaluate end-to-end QA accuracy. We evaluate retrieval metrics (nDCG, Success@k) on fixed corpora. These are different evaluation paradigms.
>
> 2. **Technical contribution:** While all three works use RL for search, our turn-level reward structure explicitly rewards backtracking and recovery (tracking rank improvements/deteriorations), enabling smaller models to learn when to pivot. This is visible in Figure 2c's backtracking analysis.
>
> We have added explicit comparison with Search-R1 and R1-Searcher in Appendix A, clarifying that our work focuses on the retrieval component while they focus on end-task performance. These are complementary directions.
>
> ### Experimental Completeness
>
> **Baseline comparisons:** We compare against:
> * **DeepRetrieval** (closest RL-based query generation method, COLM 2025) - we outperform with 2.5× smaller, domain-general models
> * **Traditional IR baselines** (BM25 variants, MiniLM embeddings)
> * **State-of-the-art LLMs** (GPT-4.1, GPT-4o, Llama 3.1-405B, Qwen3-235B)
>
> We now also have results with stronger retrievers (ModernBERT, OpenAI embeddings) showing our approach complements rather than just compensates for retriever quality. We would appreciate specific suggestions for additional baselines within our evaluation setting (offline corpora, retrieval metrics).
>
> **nDCG@10 Comparison (FEVER and SciFact)**
>
> | **Model Group** | **Model** | **FEVER** | **SciFact** |
> |-----------------|-----------|-----------|-------------|
> | **ModernBERT GTE Embeddings** | | | |
> | | Orion-Large | **0.928** | **0.802** |
> | | GPT-4.1 | 0.927 | 0.788 |
> | | Orion-Medium | 0.925 | 0.791 |
> | | Orion-Small | 0.912 | 0.776 |
> | | gte-en-base-v1.5 | 0.910 | 0.774 |
> | **OpenAI Text Embeddings** | | | |
> | | Orion-Large | **0.927** | 0.809 |
> | | GPT-4.1 | 0.920 | **0.810** |
> | | Orion-Medium | 0.921 | 0.801 |
> | | Orion-Small | 0.914 | 0.793 |
> | | text-embedding-3-large | 0.879 | 0.778 |
> | **Original MiniLM Embeddings** | | | |
> | | Orion-Large | **0.653** | **0.776** |
> | | Orion-Medium | 0.633 | 0.711 |
> | | GPT-4.1 | 0.613 | 0.724 |
> | | Orion-Small | 0.577 | 0.709 |
> | | MiniLM-L6-v2 | 0.425 | 0.505 |

---

### Official Review · Reviewer_qBuN · 2025-11-10

**Soundness:** 1
**Presentation:** 1
**Contribution:** 1
**Rating:** 2
**Confidence:** 4

**Summary:**

The authors introduce a training process to teach small language models multi-hop query generation. The training process starts with synthetic data generated to adhere to several strategies, using external LLMs, and then the authors apply step-wise GRPO to amplify useful reasoning and query chains based on retrieval metrics for each hop. They then proceed to training for future hops with a greedy selection of the highest-reward step. The authors report a number of reasonably strong results on simple, multi-hop, and reasoning-intensive retrieval tasks.

**Strengths:**

The problem motivated is reasonable. The analysis of search behaviors (Table 5) is a very interesting idea.

**Weaknesses:**

To someone who has worked in this area for several years, the writing appears almost confused or contradictory across parts of the paper. Parts of the paper seem to suggest that the emphasis is on finetuning retrieval models with LLMs via RL, other parts seem to indicate that only the reasoning LLM that generates queries is being finetuned.

To begin with, the problem of multi-hop retrieval is old; its modern instantiation is at least as old as HotPotQA (2018), a paper with over 3000 citations as there's a vast literature of methods for training what the authors refer to as "multi-turn IR" models. Indeed, many of them do things that the authors call novel, like "making the retriever itself adaptive" and "turn-level reward structure that leverages standard IR metrics to provide dense feedback at each search step". I will not cite a specific paper here because there's a vast literature of these, both through 2020-2022 with BERT-style models and through 2024-2025 (if not earlier) with more modern open LLMs and reasoning models.

That said, the main issue I take isn't about novelty. There's still plenty of room for novel methods in this space. My concern is trying to understand what the authors really did:

> The retriever itself remains static, invoked repeatedly but never trained to adapt its search strategy. This overlooks
a key point: the retrieval policy is as important as the reasoning policy. [...] We introduce a different approach: making the retriever itself adaptive. [...] A key innovation is our turn-level reward structure that leverages standard IR metrics to provide
dense feedback at each search step rather than sparse outcome-only signals.

This sounds to me like an argument to finetune the actual retriever (i.e., embedding) model. That's very reasonable, and although there's plenty of research that has done that since 2020 for multi-hop tasks, this is worth revisiting now. But the challenge is that on reading the rest of the paper we see statements like "these gains emerge not from stronger embeddings or larger scale, but from learned adaptive behavior: recognizing when queries fail, exploring alternatives systematically, and recovering from unproductive search paths." So it sounds like the authors do not, in fact, finetune the retrievers after all, and instead finetune the LLM to generate better reasoning and queries. There's an extremely large space of these types of methods, perhaps even larger than the set of methods that finetune embeddings for multi-hop retrieval.

The experiments are equally opaque. What does BM25 (dense) vs. BM25 (sparse) refer to and why is their difference in quality so drastic? What does it mean to have LLMs like GPT-4.1 in Table 2 tested against retrievers like MiniLM-L6-v2? Why only compare with the extremely weak, small, and old retrieval model MiniLM-L6-v2 from 2021 and not the many methods out there since? Many of the numerical values in Table 2 themselves are strange, when contrasted to earlier work. As an expert reader who has worked in this area for several years, I do not have confidence I understand how these results were produced.

Minor: The authors use a kind of "step-wise" GRPO for each step of the training. This is reasonable, but is there a reason not to do something like REINFORCE with intermediate rewards over the complete trajectory?

**Questions:**

Do the authors train one version of their model (at each scale) once overall or once per dataset? In other words, what is the data that the RL (GRPO) is done over?

---

> ### Author Response · Authors · 2025-11-22
>
> We sincerely thank you for your detailed feedback and the opportunity to clarify our work's positioning and contributions.
>
> ### Major Clarification: What We Train and Our Contribution
>
> **We respectfully clarify a critical misunderstanding:** We do **not** train retrieval models (embeddings). Instead, we train **small language models (350M-1.2B parameters) to generate adaptive search queries**. The retriever (MiniLM-L6-v2) remains frozen throughout. Our contribution is teaching compact models to perform iterative query reformulation with strategic backtracking, a capability traditionally requiring expensive LLM controllers.
>
> The confusion may stem from our framing around "making the retriever itself adaptive." We meant the **retrieval process** becomes adaptive through learned query generation strategies, not that we fine-tune embedding models. We have revised Section 1 and the abstract to eliminate this ambiguity.
>
> ### Positioning vs. Existing Work
>
> **Multi-hop retrieval literature:** We acknowledge the extensive literature on multi-hop IR since HotpotQA (2018). However, to the best of our knowledge, most prior work (discussed in related work) falls into three categories
> 1. **Static query decomposition/rewriting** (breaks queries upfront without corpus feedback)
> 2. **Embedding-based methods** (train better retrievers but remain one-shot)
> 3. **LLM-based agents** (use large models for adaptive search at high cost)
>
> Orion occupies a distinct position: **training compact models to internalize adaptive search policies** through RL with turn-level rewards. Our core contribution is learning to navigate the search space effectively, independent of retriever quality. Most related work targets end-to-end QA in RAG settings rather than retrieval efficacy itself.
>
> **DeepRetrieval is our closest baseline** as it also uses RL for query generation, but with key differences: (1) it trains separate 3B models per dataset versus our domain-general 1.2B models, (2) it uses PPO without SFT warm-up versus our two-stage approach, and (3) our turn-level rewards explicitly enable backtracking and recovery during search.
>
> ### Experimental Details
>
> **BM25 (dense) vs BM25 (sparse):** Thank you for catching this mislabeling. "BM25 (dense)" in Table 2 refers to results imported from [1] using E5/BGE-base embeddings, not BM25. We will either: (1) correct the label to reflect the actual dense retrieval methods (E5/BGE-base), or (2) remove these results for clarity. We use only standard BM25 in our experiments. A similar issue appears in Table 3, we will keep a single "BM25" row. We have verified all remaining baselines against prior work [1,2,3].
>
> **Why MiniLM-L6-v2?** This was a deliberate choice to create challenging conditions where learned search strategies must compensate for weak embeddings. However, **we have now validated our approach with stronger retrievers**:
>
> | **Retriever** | **Model** | **FEVER** | **SciFact** |
> |---------------|-----------|-----------|-------------|
> | ModernBERT GTE | Orion-Large | **0.928** | **0.802** |
> | | GPT-4.1 | 0.927 | 0.788 |
> | | Base retriever | 0.910 | 0.774 |
> | OpenAI | Orion-Large | **0.927** | 0.809 |
> | | GPT-4.1 | 0.920 | **0.810** |
> | | Base retriever | 0.879 | 0.778 |
> | MiniLM | Orion-Large | **0.653** | **0.776** |
> | | GPT-4.1 | 0.613 | 0.724 |
> | | Base retriever | 0.425 | 0.505 |
>
> These results confirm our strategies are **complementary to retriever quality** rather than merely compensating for weakness. Stronger embeddings + adaptive search policies yield the best results. Full results across all benchmarks will be included in the camera-ready version.
>
> **Training scope:** We train **one model per size** across all datasets (100K samples: 25K each from MS MARCO, SciFact, HotpotQA, FEVER). The model generalizes across domains without per-dataset training, a key advantage over DeepRetrieval, which trains separate 3B models per dataset. Our 1.2B model matches or exceeds DeepRetrieval while being 2.5× smaller and domain-general.
>
> **REINFORCE vs. GRPO:** Thank you for this insightful question. While vanilla REINFORCE is sound, literature documents stability concerns with large action spaces in NLP [4]. Recent work [4] shows REINFORCE works for single-turn LLM alignment with strong initialization, but for our small models untrained for search, GRPO provided: (1) reduced variance through group-based baselines, (2) lower memory usage, (3) stable multi-turn training dynamics. Our choice was implementation-driven rather than fundamental rejection of simpler methods. Recent REINFORCE++ [5] is compelling, but computational constraints prevented comparison. We view this as valuable future work.
>
> Given these clarifications, would you consider raising your score for our paper?
>
> ---
>
> **References:**
> [1] Jiang et al., DeepRetrieval, COLM 2025
> [2] Su et al., BRIGHT, arXiv 2024
> [3] Das et al., RADER, EMNLP 2025
> [4] Ahmadian et al., Back to Basics, arXiv 2024
> [5] Hu et al., REINFORCE++, arXiv 2025

---

### Meta-Review · Area_Chair_N1i4 · 2026-01-07

**Summary:**

This paper proposes Orion, a training framework enabling small language models to perform adaptive multi-turn retrieval through learned search strategies using synthetic trajectory generation, SFT, and RL with turn-level rewards. While the paper demonstrates strong empirical results, all four reviewers did not recommend acceptance due to concerns about unclear novelty relative to existing multi-hop retrieval literature and recent works, missing critical baseline comparisons, weak RL contribution justification, and fundamental confusion about what models are being trained (or more broadly, paper clarity). The rebuttal addresses some clarity issues and provides new experiments with stronger retrievers, but leaves core concerns about novelty, missing comparisons, and methodology presentation unresolved.

**Reviewer Concerns:**

Critical concerns remain unresolved: (1) novelty: the rebuttal distinguishes from Search-R1/R1-Searcher based on evaluation paradigm (retrieval metrics vs QA) rather than fundamental technical contributions, which do not seem sufficient. No specific citations or detailed comparison with the vast multi-hop retrieval literature. (2) missing comparisons: No experimental results comparing with Search-R1/R1-Searcher baselines despite their direct relevance. DeepRetrieval comparison remains incomplete on most benchmarks. Modern retrieval methods (ColBERTv2, RankRAG, hybrid pipelines) still not tested. (3) RL justification: GRPO adds only 1-2% over SFT with no ablation comparing against PPO/DPO/REINFORCE. (4) Serious presentation issues remain.

**Reviewer Scores:**

Originally, all reviewers recommended reject. Given that the rebuttal didn't address major concerns, it is expected that the final scores will remain pretty much the same.

---

### Decision · Program_Chairs · 2026-01-26

Reject